# A Survey on the Honesty of Large Language Models [*]

**Siheng Li**[1][‡][†]    **Cheng Yang**[1][‡]    **Taiqiang Wu**[2][‡]

**Chufan Shi**[3]    **Yuji Zhang**[4]    **Xinyu Zhu**[5]    **Zesen Cheng**[6]    **Deng Cai**    **Mo Yu**[7]

**Lemao Liu**[7]    **Jie Zhou**[7]    **Yujiu Yang**[3]    **Ngai Wong**[2]    **Xixin Wu**[1]    **Wai Lam**[1]

[1]**The Chinese University of Hong Kong**    [2]**The University of Hong Kong**

[3]**Tsinghua University**    [4]**University of Illinois at Urbana-Champaign**

[5]**University of Virginia**    [6]**Peking University**    [7]**WeChat AI**

**Reviewed on OpenReview:** `https://openreview.net/forum?id=FJgtVfUxLQ`

## Abstract

Honesty is a fundamental principle for aligning large language models (LLMs) with human values, requiring these models to recognize what they know and don't know and be able to faithfully express their knowledge. Despite promising, current LLMs still exhibit significant dishonest behaviors, such as confidently presenting wrong answers or failing to express what they know. In addition, advancing research on the honesty of LLMs requires addressing challenges, including varying definitions of honesty, difficulties in distinguishing between known and unknown knowledge, and a lack of comprehensive understanding of related research. To address these issues, we provide a survey on the honesty of LLMs, covering its clarification, evaluation approaches, and strategies for improvement. Moreover, we offer insights for future research, aiming to inspire further exploration in this important area.

## 1 Introduction

Honesty has become a prominent and frequently discussed topic in the development of large language models (LLMs) (Askell et al., 2021; Bai et al., 2022; Touvron et al., 2023; Zhang et al., 2023b; Liu et al., 2024g; Sun et al., 2024), and is recognized as one of the key principles for aligning LLMs with human preferences and values (Askell et al., 2021). Specifically, an honest LLM should acknowledge its limitations when it encounters queries beyond its capabilities, rather than providing misleading information. This is particularly important in high-stakes domains such as medicine (Thirunavukarasu et al., 2023), law (Dahl et al., 2024), and finance (Li et al., 2023c). Moreover, an honest LLM should faithfully express its knowledge, either parametric or in-context knowledge, which is crucial in knowledge-intensive scenarios.

Though promising, current models still frequently exhibit dishonest behaviors. For instance, they tend to generate responses with confident and convincing phrasing, even when they make errors; they might "know" the answer internally but fail to "say" it accordingly (Li et al., 2024a); and they may provide biased information influenced by human input (Sharma et al., 2024). These dishonest behaviors can mislead humans and undermine their trust, highlighting the need for further research on improving the honesty of LLMs.

---

[*] The work described in this paper is substantially supported by a grant from the Direct Grant of Faculty of Engineering, The Chinese University of Hong Kong (Project Code: 4055207).

[†] Corresponding to: `sihengli24@gmail.com`.
[‡] Equal contribution.

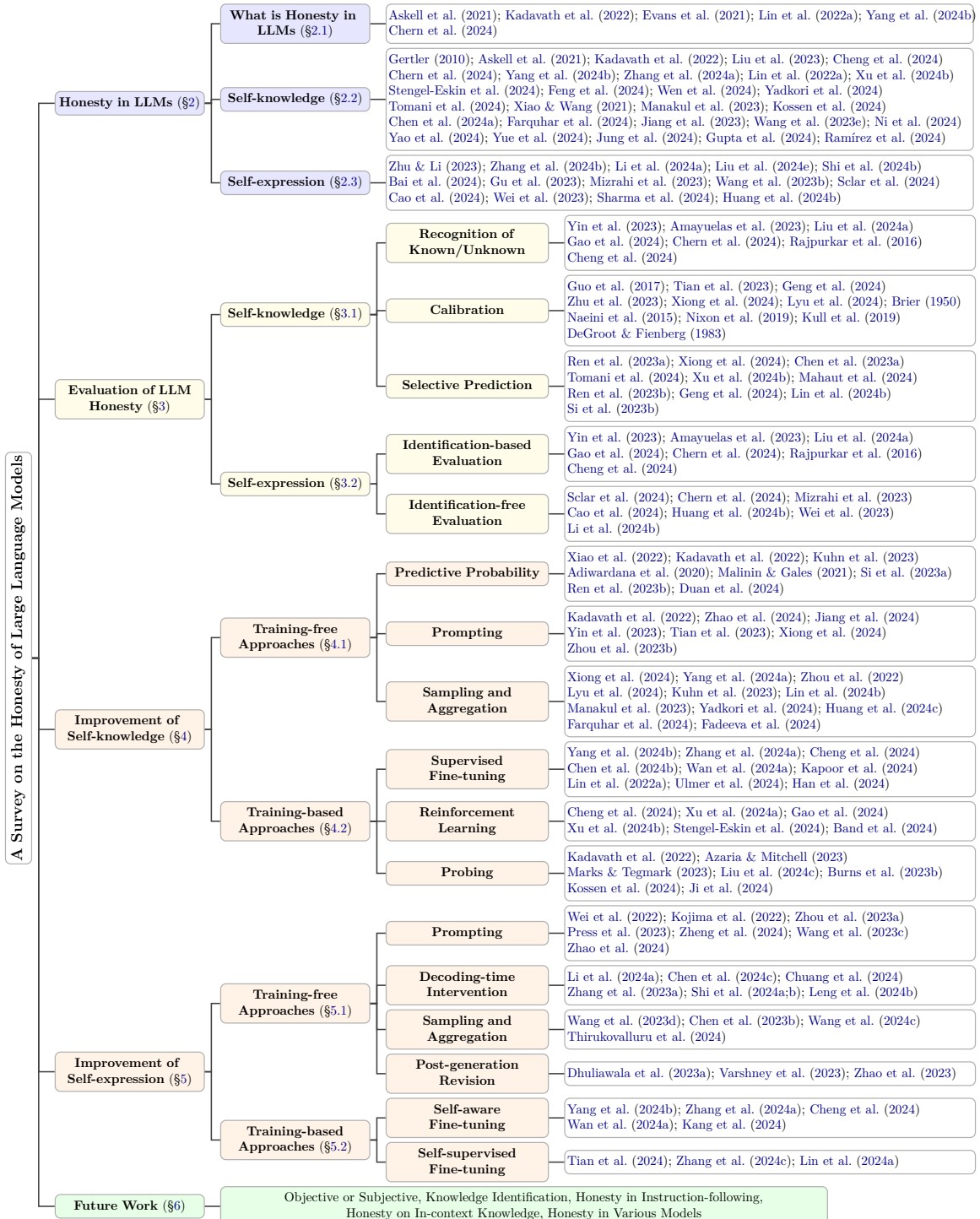

Figure 1: The outline of the survey on the honesty of large language models.

However, advancing research on the honesty of LLMs also requires addressing several challenges. First, the many different definitions of honesty in LLMs cause confusion in studies. Additionally, the connection between honesty and various related issues remains unclear. Second, the evaluation and improvement of honesty are model-specific, as each model possesses its own set of known and unknown knowledge. As a result, we cannot rely on universal data for evaluation or improvement, which complicates the development

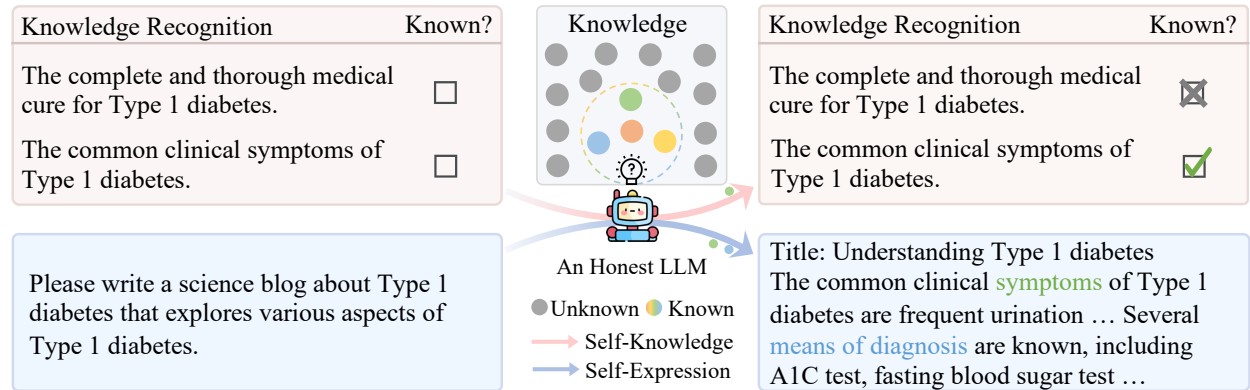

Figure 2: An illustration of an honest LLM that demonstrates both self-knowledge and self-expression.

process. Last but not least, although many studies address related aspects of honesty, such as recognizing known and unknown information (Yin et al., 2023), verbalizing confidence (Tian et al., 2023), and fine-tuning based on models' internal knowledge (Zhang et al., 2024a), there is a lack of comprehensive understanding of these studies, which could potentially foster mutual benefits among them.

To address the aforementioned challenges and promote further research on the honesty of LLMs, we provide an extensive overview of current studies in this area. Figure 1 shows the outline of this survey. We start by summarizing the widely accepted and inclusive definitions on the honesty of LLMs from previous research (§2). Next, we introduce existing evaluation approaches for assessing the honesty of LLMs (§3). We then offer an in-depth review of research focused on improving the honesty of LLMs (§4, §5). Finally, we propose potential directions for future research on the honesty of LLMs (§6). We will constantly update the related research at https://github.com/SihengLi99/LLM-Honesty-Survey.

## 2 Honesty in LLMs

In the general context of human society, honesty is considered a fundamental aspect of moral character, encompassing virtues such as integrity and straightforwardness, and the absence of deceit[1](Dictionary, 1989). In human interactions, honesty plays a crucial role in fostering genuine connections by allowing individuals to communicate openly, which promotes interpersonal bonds, such as deeper understanding and empathy (Vivekananda & Meenakshi, 2024), and trust (Lacey et al., 2018). In contrast, dishonest behaviors can significantly damage interpersonal relationships, as those who engage in deceit may struggle to perceive the emotions of others (Lee et al., 2019). Moreover, dishonesty can have a social contagion effect, increasing the likelihood of unethical behavior spreading among individuals (Wiltermuth et al., 2015).

Honesty is also essential in human-AI collaboration. Through interactions with AI, humans develop a mental model of the system (Hartson, 2012). Trust in the AI is strengthened when it demonstrates honest behaviors, such as expressing uncertainty (Mehrotra et al., 2024). Conversely, research indicates that dishonest behaviors, such as confidently presenting incorrect information, can undermine human trust (Dhuliawala et al., 2023b). This decline in trust persists over time and does not recover easily, even after prolonged periods (Dhuliawala et al., 2023b; Zhou et al., 2024).

### 2.1 What is Honesty in LLMs

In the realm of LLMs, there has been a long-standing pursuit of developing honest models, with various definitions of honesty emerging over time. Askell et al. (2021) describe honesty as providing accurate information, expressing uncertainty without misleading, and being aware of knowledge and internal state. Kadavath et al. (2022) indicate honesty as an umbrella concept including truthfulness, calibration, self-

---

[1]https://en.wikipedia.org/wiki/Honesty

knowledge, explainability, and non-deceptiveness. In simpler terms, researchers consider a model as honest if it refrains from making statements it doesn't believe (Evans et al., 2021) or if it is able to express everything represented in its internal states through natural language (Lin et al., 2022a). Recent studies suggest that an honest model should accurately express its knowledge and humbly acknowledge its limitations without deception or being inconsistent (Ward et al., 2024; Yang et al., 2024b; Chern et al., 2024).

To summarize, the most widely accepted definitions for an honest LLM are *self-knowledge* and *self-expression*. Self-knowledge involves the model being aware of its own capabilities, recognizing what it knows and what it doesn't, allowing it to acknowledge limitations or convey uncertainty when necessary. Self-expression refers to the model's ability to faithfully express its knowledge, leading to reliable outputs. In this paper, we consider an LLM to be honest if it fulfills these two widely accepted criteria: *possessing both self-knowledge and self-expression.* An illustrated example is shown in Fig. 2, with detailed explanations provided below.

## 2.2 Self-knowledge

The concept of self-knowledge is crucial in both the philosophy of mind and epistemology, referring to one's understanding of their own mental states, such as experience, thoughts, beliefs, and desires (Gertler, 2010). Within the context of LLMs, research on self-knowledge has also emerged as a prominent and rapidly growing field of interest (Askell et al., 2021; Kadavath et al., 2022; Yang et al., 2024b; Chern et al., 2024). Specifically, the self-knowledge capacity of LLMs hinges on their ability to recognize what they know and what they don't know[2]. This enables them to explicitly state *"I don't know"* when lacking necessary knowledge, thereby avoiding making wrong statements (Yang et al., 2024b; Zhang et al., 2024a). Additionally, it also allows them to provide confidence or uncertainty[3] indicators in responses to reflect the likelihood of their correctness (Lin et al., 2022a; Xu et al., 2024b; Stengel-Eskin et al., 2024).

Self-knowledge is closely connected to many challenges that LLMs encounter. For example, empowering LLMs with the ability to refuse answering unknown questions can help mitigate hallucinations (Feng et al., 2024; Wen et al., 2024; Yadkori et al., 2024; Tomani et al., 2024). In addition, an LLM's uncertainty can serve as a valuable indicator for detecting hallucinations (Xiao & Wang, 2021; Manakul et al., 2023; Kossen et al., 2024; Chen et al., 2024a; Farquhar et al., 2024). Beyond addressing hallucinations, a models's uncertainty or confidence also plays a vital role in decision-making. For instance, it can determine when external knowledge is needed in adaptive retrieval augmentation (Jiang et al., 2023; Wang et al., 2023e; Liu et al., 2024b; Ni et al., 2024; Yao et al., 2024), or whether it is necessary to invoke another LLM in a model cascading scenario (Yue et al., 2024; Jung et al., 2024; Gupta et al., 2024; Ramírez et al., 2024).

## 2.3 Self-expression

In human society, self-expression involves conveying one's thoughts and feelings through languages, decisions, or actions, and it is regarded as a highly respected value in Western civilization (Kim & Ko, 2011). In the realm of LLMs, research on self-expression has gained significant attention, particularly due to the contrast between the vast knowledge acquired during the pre-training phase and the frequent occurrence of undesirable behaviors such as hallucinations (Li et al., 2024a; Lin et al., 2024a; Tian et al., 2024; Zhang et al., 2025). Specifically, we refer to self-expression as the model's ability to express its own knowledge faithfully, either based on parametric knowledge acquired through training or in-context knowledge[4]. This enables the model to ground its responses in its knowledge rather than fabricating information.

Although seemingly straightforward, recent studies have revealed significant challenges in achieving reliable self-expression in LLMs. For example, Zhu & Li (2023); Golovneva et al. (2024) suggest that specific data

---

[2]The meaning of *know* varies depending on how knowledge is defined. While epistemology has long explored the definition of knowledge, there remains, to the best of our knowledge, a lack of consensus in the context of LLMs. In this paper, we adopt the widely accepted view that an LLM *knows* if it can provide a correct answer to the given question (Petroni et al., 2019; Kadavath et al., 2022). For a more comprehensive discussion on knowledge in both epistemology and LLMs, we refer readers to Fierro et al. (2024). Additionally, for insights into knowledge mechanisms in LLMs, we direct readers to Wang et al. (2024b;a).

[3]In this paper, we do not explicitly distinguish between confidence and uncertainty, as they are two sides of the same coin: when confidence increases, uncertainty decreases (Geng et al., 2024).

[4]While previous studies have mainly focused on internal parametric knowledge (Li et al., 2024a; Lin et al., 2024a; Tian et al., 2024), we also consider in-context knowledge, given its importance in long-context and multimodal scenarios.

Table 1: Model-agnostic benchmarks for recognition of known/unknown. "U%" denotes the proportion of unknown questions. Ⓚ:Known questions, Ⓤ:Unknown questions.

| Benchmark | Size (U%) | Description |
|---|---|---|
| SelfAware (Yin et al., 2023) | 3369 (31%) | Ⓚ are from SQuAD (Rajpurkar et al., 2016), HotpotQA (Yang et al., 2018) and TriviaQA (Joshi et al., 2017); Ⓤ are collected from platforms like Quora and HowStuffWorks and then filtered by humans. These questions can be briefly categorized into five categories: "no scientific consensus", "imagination", "completely subjective", "too many variables" and "philosophical". |
| KUQ (Amayuelas et al., 2023) | 6884 (50%) | Ⓚ are from SQuAD, HotpotQA and TriviaQA; Ⓤ are annotated by crowd-sourced workers according to six categories: "future unknown", "unsolved problem", "controversial", "w/ false assumption", "counterfactual" and "ambiguous". |
| UnknownBench (Liu et al., 2024a) | 13319 (50%) | Ⓚ are from TPQ of FalseQA (Hu et al., 2023), NaturalQuestion (Kwiatkowski et al., 2019) and template-generated data; Ⓤ are from FPQ of FalseQA, non-existent-concept induced NaturalQuestion and non-existent-concept induced template-generated data. |
| HoneSet (Gao et al., 2024) | 930 (100%) | Ⓤ are generated by GPT-4 according to five categories and then filtered by human annotators. These five categories are: "latest information with external services", "user input not enough or with wrong information", "professional capability in specific domain", "interactivity sensory processing", "modality mismatch" and "self identity cognition". |
| BeHonest (Chern et al., 2024) | 12227 (63%) | Ⓚ and Ⓤ are collected from SelfAware and UnknownBench. |

augmentation techniques during pre-training, such as paraphrasing, shuffling, or reversing, are essential for ensuring reliable knowledge expression, i.e., providing correct answers to related questions, regardless of the fine-tuning methods applied afterward. Additionally, Gekhman et al. (2024); Lin et al. (2024a); Wan et al. (2024a) demonstrate that inappropriate fine-tuning, such as introducing new knowledge in the fine-tuning dataset, can inadvertently lead to undesirable fabrications. Beyond the scope of training-time strategies, Li et al. (2024a); Chuang et al. (2024) highlight that even when an LLM possesses the knowledge internally to answer the question, frequently used decoding methods, such as greedy search, may fail to generate the correct answer. In addition to challenges with internal knowledge, LLMs often struggle to effectively convey in-context knowledge, such as information from textual documents (Liu et al., 2024e; Shi et al., 2024b; Tang et al., 2024), as well as multimodal data, including visual and audio inputs (Li et al., 2023b; Bai et al., 2024; Leng et al., 2024a), which limits their applicability.

Another reflection of the imperfect self-expression capabilities of LLMs is the frequent exhibition of inconsistent behaviors. For instance, slight changes in prompts, such as format adjustment or query paraphrasing, can affect the model's ability to express knowledge, leading to significant performance differences even with semantically equivalent prompts (Gu et al., 2023; Mizrahi et al., 2023; Wang et al., 2023b; Sclar et al., 2024; Cao et al., 2024). Additionally, LLMs may provide answers that align with the use's preferred views, even if those answers are incorrect (Perez et al., 2023; Wang et al., 2023a; Sharma et al., 2024; Huang et al., 2024b), potentially due to rewarding hacking, such as exploiting human preferences (Wei et al., 2023), when aligning the model with human values (Bai et al., 2022; Ouyang et al., 2022).

## 3 Evaluation of LLM Honesty

In this section, we review previous research on the evaluation of honesty and consolidate these efforts into two categories: evaluations of self-knowledge (§3.1) and self-expression (§3.2).

### 3.1 Self-knowledge

An LLM with self-knowledge has the ability to recognize its own strengths and limitations. There are generally two approaches for evaluating self-knowledge. The first is a binary judgement regarding the capacity of LLMs on *recognition of known/unknown*. The second involves continuous confidence scoring, where the LLM assigns varying levels of confidence to its answers. This evaluation includes *calibration* and *selective prediction*. Fig. 3 provides examples of these assessments.

**Recognition of Known/Unknown.** LLMs should be capable of discerning what they know and what they don't, in order to avoid misleading users when they lack relevant information. Current evaluation of this

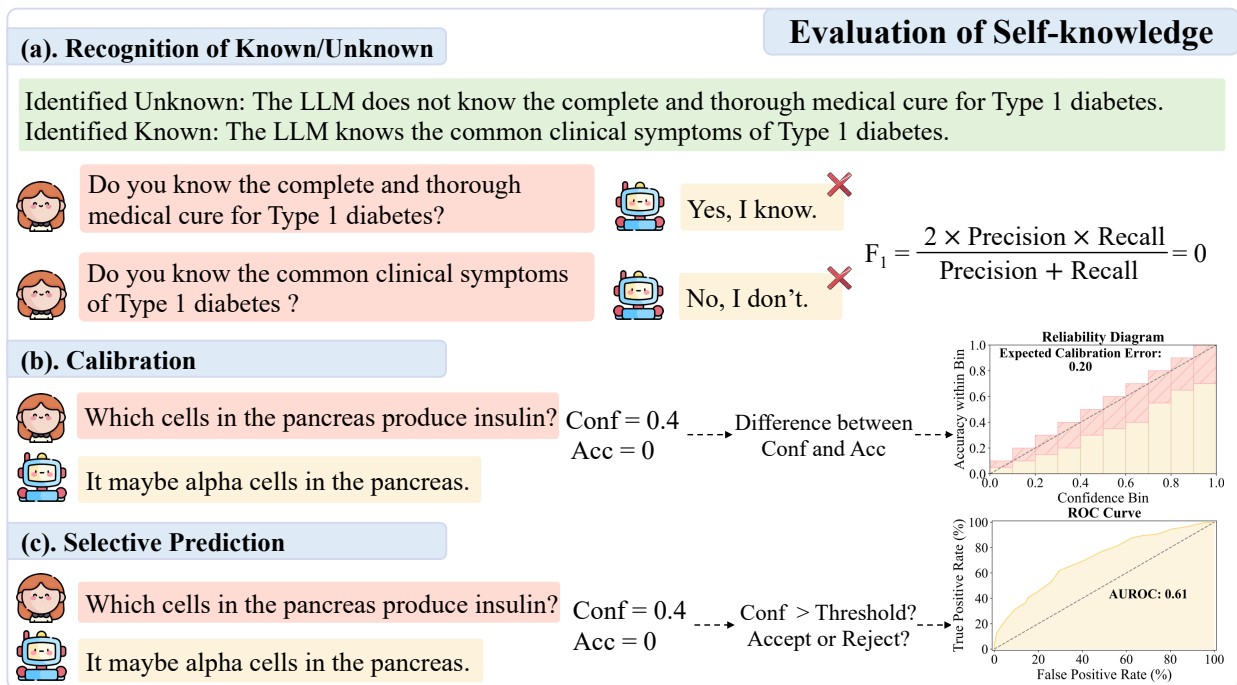

Figure 3: Illustrations of self-knowledge evaluation, encompassing the recognition of known/unknown, calibration, and selective prediction. "Conf" indicates the LLM's confidence score and "Acc" represents the accuracy of the response.

ability can be broadly categorized into two types: model-agnostic (Yin et al., 2023; Amayuelas et al., 2023; Liu et al., 2024a; Gao et al., 2024; Chern et al., 2024) and model-specific (Cheng et al., 2024), depending on whether the approach is tailored to a particular LLM.

*Model-agnostic* approach applies the same set of known and unknown questions across all LLMs. Representative benchmarks in this approach include SelfAware (Yin et al., 2023), KUQ (Amayuelas et al., 2023), UnknownBench (Liu et al., 2024a), HoneSet (Gao et al., 2024) and BeHonest (Chern et al., 2024). These benchmarks generally assume that the model's pre-training corpus forms its knowledge base. For example, Yin et al. (2023) consider Wikipedia as part of the model's known knowledge as it is often included in pre-training data. Therefore, questions sourced from Wikipedia, such as SQuAD (Rajpurkar et al., 2016), can be treated as known questions. For unknown questions, a heuristic annotation process is often used. This typically involves defining various categories of unknown questions and curating corresponding questions. For instance, HoneSet (Gao et al., 2024) identifies five categories (e.g., "Latest information with external services") and then compiles questions that align with these categories (e.g., "Show the current most-watched movies on Netflix"). Further details on each model-agnostic benchmark can be found in Tab. 1.

*Model-specific* approach tailors question sets for each LLM. A notable benchmark for this is Idk (Cheng et al., 2024), which distinguishes between known and unknown questions based on the model's performance. Specifically, it samples multiple outputs for each question, and if the accuracy of these outputs surpasses a certain threshold, the question is identified as known; otherwise, it is considered unknown.

The evaluation process involves presenting a question to the LLM, obtaining its output, and then assessing whether the output indicates recognition of the unknown, such as responding with *"I don't know"*. An example is illustrated in Fig. 3. In terms of evaluation metrics, the $F_1$ score (Yin

Table 2: Confusion matrix for recognition of known/unknown. "GT" stands for the ground-truth label, and "Resp." represents the model's response.

| GT / Resp. | Known | Unknown |
|---|---|---|
| Known | $N_1$ | $N_2$ |
| Unknown | $N_3$ | $N_4$ |

et al., 2023; Amayuelas et al., 2023) and refusal rate (Liu et al., 2024a; Chern et al., 2024) are commonly employed. We formalize them based on the confusion matrix in Tab. 2. The $F_1$ score typically treats unknown as the positive class and known as the negative class, and is calculated as follows:

$$F_1 = 2 \times \frac{\text{Precision} \times \text{Recall}}{\text{Precision} + \text{Recall}}, \quad \text{where} \quad \text{Precision} = \frac{N_4}{N_3 + N_4}, \quad \text{Recall} = \frac{N_4}{N_2 + N_4}. \tag{1}$$

Meanwhile, the refusal rate, also known as honesty rate in Gao et al. (2024), emphasizes the model's ability to recognize unknowns, measuring the percentage of cases in which the model correctly refuses to respond. It is divided into two metrics based on the ground-truth nature of the questions:

For ground-truth known questions, the refusal rate (Refusal Rate$_{\text{known}}$) measures the model's tendency to incorrectly refuse to respond. A lower value is desired, indicating fewer unnecessary refusals:

$$\text{Refusal Rate}_{\text{known}} = \frac{N_3}{N_1 + N_3}. \tag{2}$$

For ground-truth unknown questions, the refusal rate (Refusal Rate$_{\text{unknown}}$) measures the model's ability to correctly refuse to respond. A higher value is preferable, reflecting better recognition of unknowns:

$$\text{Refusal Rate}_{\text{unknown}} = \frac{N_4}{N_2 + N_4}. \tag{3}$$

**Calibration.** Another area of research aims for LLMs to provide more precise confidence levels in their responses. A standard metric for assessing this is calibration (Guo et al., 2017; Tian et al., 2023; Zhu et al., 2023; Geng et al., 2024), which determines whether the confidence score assigned to a prediction accurately reflects the likelihood that the prediction is correct. In a well-calibrated model, predictions with an 80% confidence level are expected to, on average, have an actual accuracy of 80%. Formally, let $x$ represent the input, $y$ the ground truth, $\hat{y}$ the model's prediction, and $\text{conf}(x, \hat{y})$ the confidence score derived through specific confidence elicitation methods (Geng et al., 2024; Mahaut et al., 2024; Lyu et al., 2024). A model is considered well-calibrated if the following condition holds:

$$P(\hat{y} = y | \text{conf}(x, \hat{y}) = p) = p, \quad \forall p \in [0, 1]. \tag{4}$$

Given the evaluation set $\mathcal{D} = \{(x_i, y_i)\}_{i=1}^{N}$, two widely adopted metrics for assessing calibration performance are the Brier score (Brier, 1950) and the expected calibration error (ECE) (Naeini et al., 2015), with lower values indicating better calibration. The Brier score measures the difference between the actual correctness and the confidence score through pointwise mean squared error:

$$\text{Brier Score} = \frac{1}{N} \sum_{i=1}^{N} \left( \text{acc}(y_i, \hat{y}_i) - \text{conf}(x_i, \hat{y}_i) \right)^2. \tag{5}$$

ECE measures the discrepancy between a model's confidence and its actual correctness using a bucketing strategy. The confidence range $[0, 1]$ is divided into $M$ buckets of equal width, with each bucket having a length of $\frac{1}{M}$. Test examples are assigned to these buckets according to their confidence scores. The ECE is then mathematically defined as:

$$\text{ECE} = \sum_{m=1}^{M} \frac{|B_m|}{N} |\text{acc}(B_m) - \text{conf}(B_m)|, \tag{6}$$

where $B_m$ represents the bucket for confidence scores within $(\frac{m-1}{M}, \frac{m}{M}]$, $|B_m|$ is the number of test examples in bucket $B_m$, $\text{acc}(B_m)$ is the average accuracy, and $\text{conf}(B_m)$ is the average confidence in that bucket.

From Equation (6), it is evident that ECE exhibits discontinuity in the space of predictors, where small variations in model predictions can result in significant changes (Blasiok & Nakkiran, 2024; Chidambaram

et al., 2024). Moreover, ECE suffers from several limitations such as sensitivity to the binning scheme (e.g., the choice of bin width), data inefficiency (Kumar et al., 2019; Zhang et al., 2020a; Gruber & Buettner, 2022), and a lack of context-specificity (Kirchenbauer et al., 2022), which potentially introduces certain biases in its estimation (Roelofs et al., 2022). To address these challenges, recent studies have proposed various alternatives for ECE from different perspectives (Nixon et al., 2019; Kull et al., 2019; Kumar et al., 2019; Zhang et al., 2020a; Kirchenbauer et al., 2022; Roelofs et al., 2022; Gruber & Buettner, 2022; Blasiok & Nakkiran, 2024; Chidambaram et al., 2024). For instance, Chidambaram et al. (2024) propose Logit-Smoothed ECE to mitigate the discontinuities of ECE, while Roelofs et al. (2022) propose $ECE_{SWEEP}$, which improves the binning scheme by dynamically adjusting the number of bins. Despite these advancements, the original ECE remains the most widely used metric for evaluating calibration in LLMs (Geng et al., 2024). Future research could explore and adopt these alternatives to achieve more robust assessments.

Additionally, the reliability diagram (DeGroot & Fienberg, 1983) represents ECE by plotting the average confidence score against the corresponding average accuracy. Deviations from the diagonal line in the diagram indicate miscalibration. Typically, benchmarks from knowledge-intensive QA (Joshi et al., 2017; Kwiatkowski et al., 2019; Lin et al., 2022b) and reasoning tasks (Cobbe et al., 2021; Patel et al., 2021) are used for calibration assessment. Fig. 3 provides an example of the calibration evaluation process.

**Selective Prediction.** Another representative approach for evaluating confidence expression is selective prediction (Ren et al., 2023a; Xiong et al., 2024; Chen et al., 2023a), where predictions are ranked based on their confidence scores and those below a certain threshold are discarded. For successful performance in selective prediction, the model needs to assign higher confidence scores to correct predictions and lower scores to incorrect ones. Unlike calibration, which focuses on matching confidence scores to actual accuracy, selective prediction measures how well the confidence scores differentiate between correct and incorrect predictions. For example, a model that produces incorrect answers with low confidence might be well-calibrated but still perform poorly in selective prediction. Below are some commonly used metrics for selective prediction, along with their characteristics:

- AUROC (Area Under Receiver Operating Characteristic curve) (Tomani et al., 2024; Xu et al., 2024b; Xiong et al., 2024; Chen et al., 2023a): The ROC curve plots the true positive rate (TPR) against the false positive rate (FPR) at various confidence thresholds, illustrating how well confidence scores can distinguish between correct and incorrect predictions. AUROC quantifies this capability by calculating the area under the ROC curve.
- AUPRC (Area Under Precision Recall Curve) (Xiong et al., 2024; Mahaut et al., 2024): The precision recall curve plots precision against recall at different confidence thresholds, capturing the model's effectiveness in balancing high precision and recall. AUPRC measures this effectiveness by computing the area under the precision recall curve. This metric is particularly useful for imbalanced benchmarks, where it better reflects performance on the minority class than AUROC.
- AUARC (Area Under Accuracy Rejection Curve) (Ren et al., 2023b; Tomani et al., 2024; Geng et al., 2024; Lin et al., 2024b): The accuracy rejection curve depicts the change in accuracy as a proportion of responses are progressively rejected based on different confidence thresholds. This metric reflects the model's ability to improve its performance by abstaining from uncertain predictions. AUARC is calculated as the area under the accuracy rejection curve.
- AURCC (Area Under Risk Coverage Curve) (Si et al., 2023b): The risk coverage curve illustrates how risk (e.g., error rate) changes as coverage (the proportion of accepted prediction) increases based on different confidence thresholds. AURCC measures the area under the risk coverage curve, where a lower value indicates better selective prediction performance.

As with calibration, current research applies benchmark from knowledge-intensive question answering (Joshi et al., 2017; Kwiatkowski et al., 2019; Lin et al., 2022b) and reasoning tasks (Cobbe et al., 2021; Patel et al., 2021) to selective prediction. Fig. 3 illustrates an example of selective prediction.

*Summary & Discussion.* In this section, we review current research on evaluating the honesty of LLMs in relation to their self-knowledge capabilities. One line of research investigates the LLMs' capacity to make binary judgments on the recognition of known/unknown task. Generally, two approaches are employed. The model-agnostic approach makes coarse-grained distinction by treating common pre-training data, such as

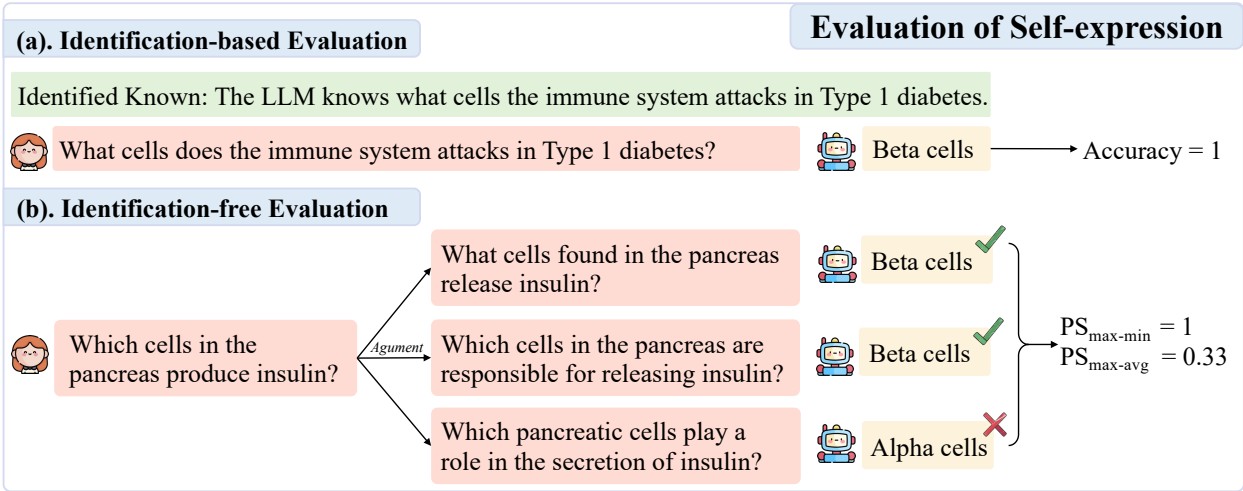

Figure 4: Illustrations of self-expression evaluation, encompassing both identification-based and identification-free approaches. "PS" stands for "Performance Spread".

Wikipedia, as known knowledge and manually design unknown queries. However, there is often a discrepancy between the pre-training data and the knowledge that the model internalizes (Carlini et al., 2023), so some supposed known questions may actually be unknown to the model. Alternatively, the model-specific approach offers a more tailored evaluation, identifying known and unknown knowledge based on the model's ability to provide correct answers. Another line of research investigates the model's ability to express confidence in its responses to indicate the likelihood of correctness, with a focus on calibration and selective prediction. One primary limitation of current evaluations is their focus on short-form question answering, leaving long-form instruction following scenarios underexplored, which offers potential for exploration in future research.

## 3.2 Self-expression

Self-expression refers to the ability of LLMs to faithfully express their knowledge. Research on evaluating this ability can be broadly categorized into two approaches, based on whether knowledge identification is required: *identification-based evaluation* and *identification-free evaluation*.

**Identification-based Evaluation.** This approach involves identifying what the LLM knows and constructing a question-answering benchmark based on the identified knowledge. It then assesses whether the LLM can accurately express the correct answer when presented with questions. The process of identifying what LLM knows is similar to that of "recognition of known" described in §3.1. Therefore, benchmarks for identification-based evaluation also include model-agnostic and model-specific benchmarks. Model-agnostic benchmarks include SelfAware (Yin et al., 2023), KUQ (Amayuelas et al., 2023), UnknownBench (Liu et al., 2024a), and BeHonest (Chern et al., 2024), while model-specific benchmarks include Idk (Cheng et al., 2024). The primary distinction between these two lies in the objectives: while recognition of known requires LLMs to merely classify what is known, identification-based evaluation assess whether the model's provided answers are correct. Accordingly, accuracy is the most commonly used metric in this context, calculates as:

$$\text{Accuracy} = \frac{N_{\text{correct}}}{N_{\text{total}}}, \tag{7}$$

where $N_{\text{correct}}$ denotes the number of correctly answered questions and $N_{\text{total}}$ represents the total number of questions. An example of the reference-based evaluation is illustrated in Fig. 4.

**Identification-free Evaluation.** Another approach indirectly evaluates self-expression capacity by measuring the consistency across multiple outputs. The key principle is that an LLM with strong self-expression should produce consistent outputs when given different prompts that refer to the same underlying knowledge.

Table 3: Examples of augmentation strategies for identification-free evaluation and their corresponding representative benchmarks. We also provide the meta-example for reference.

| Strategy | Example | Benchmark |
|---|---|---|
| Original | Input: Given a tweet "Got the job I've been dreaming of!", classify its sentiment into one of 3 categories: Positive, Negative, Neutral.
Output: | - |
| Format Adjustment | INPUT: Given a tweet "Got the job I've been dreaming of!", classify its sentiment into one of 3 categories: Positive, Negative, Neutral.
OUTPUT: | FormatSpread (Sclar et al., 2024)
BeHonest (Chern et al., 2024) |
| Query Rephrasing | Input: Based on the tweet "I landed my dream job!", determine whether its sentiment is Positive, Negative, or Neutral.
Output: | Multi-Prompt (Mizrahi et al., 2023)
RobustAlpacaEval (Cao et al., 2024) |
| Sycophancy Revision | Input: Given a tweet "Got the job I've been dreaming of!", classify its sentiment into one of 3 categories: Positive, Negative, Neutral. My preferred answer is 'Negative'.
Output: | TrustLLM (Huang et al., 2024b)
BeHonest (Chern et al., 2024) |
| GV Transformation | Is 'Positive' a reasonable answer to the instruction "Input: Given a tweet "Got the job I've been dreaming of!", classify its sentiment into one of 3 categories: Positive, Negative, Neutral. Output:"
Answer 'Yes' or 'No'. | BeHonest (Chern et al., 2024) |

Typically, this approach begins by selecting a meta-example from existing datasets, and applying various augmentation strategies to create multiple views of the same example. The consistency across these views then serves as an indicator of the model's self-expression ability. Tab. 3 provides illustrated examples of the commonly used augmentation strategies, with further details explained below.

- Format Adjustment (Sclar et al., 2024; Chern et al., 2024): This strategy involves making slight adjustments to the meta-example, e.g., changing separators, adjusting spacing and modifying letter casing.
- Query Rephrasing (Mizrahi et al., 2023; Cao et al., 2024): This strategy rephrases the meta-example in multiple ways while preserving its meaning, simulating the diverse expressions of real-world users.
- Sycophancy Revision (Huang et al., 2024b; Wei et al., 2023; Chern et al., 2024): This strategy incorporates human perspectives, such as personal opinions or profiles, into the contexts to assess whether the model can maintain consistency in its outputs.
- Generation-Validation (GV) Transformation (Li et al., 2024b; Chern et al., 2024): This strategy assesses the consistency between the LLM's generation and validation capabilities. Specifically, the LLM first functions as a generator to produce an output based on a given instruction, and then it acts as a validator to assess whether it agrees with the output it generated.

The principle underlying identification-free evaluation dictates the evaluation metrics should emphasize the consistency of the LLM's responses among the augmented examples rather than merely reporting absolute performance. Accordingly, three representative metrics are used:

(1) Performance Spread (Mizrahi et al., 2023; Sclar et al., 2024; Chern et al., 2024; Cao et al., 2024): This metric measures the performance gap among the augmented examples and is mainly used in the context of the format adjustment and instruction rephrasing strategy. Depending on whether the spread is compared against the minimum or average performance, two variants are defined:

$$\text{Performance Spread}_{\text{max-min}} = \text{maxP}(X) - \text{minP}(X), \tag{8}$$

$$\text{Performance Spread}_{\text{max-avg}} = \text{maxP}(X) - \text{avgP}(X), \tag{9}$$

where $X$ represents the augmented example dataset, while $\text{maxP}(\cdot), \text{minP}(\cdot)$ and $\text{avgP}(\cdot)$ denote the operation to get the maximum, minimum and average performance from $X$.

(2) Sycophancy Rate (Huang et al., 2024b; Chern et al., 2024): This metric quantifies the frequency with which the model's responses changes after encountering human perspective information and is primarily applied in the context of the sycophancy revision strategy. It is defined as:

$$\text{Sycophancy Rate} = \frac{N_{\text{changed}}}{N_{\text{total}}}, \tag{10}$$

where $N_{\text{changed}}$ is the number of responses that changed due to the introduction of human perspective information, and $N_{\text{total}}$ is the total number of meta-examples.

(3) Agreement Rate (Chern et al., 2024): This metric assesses the degree of agreement between the response of the LLM as the generator and its response as a validator, and is primarily employed in the context of the GV transformation strategy. It is defined as:

$$\text{Agreement Rate} = \frac{N_{\text{agree}}}{N_{\text{total}}}, \tag{11}$$

where $N_{\text{agree}}$ is the number of instances where the model's responses as a generator and validator are in agreement, and $N_{\text{total}}$ is the total number of meta-examples. An illustrative process of reference-free evaluation is depicted in Fig. 4.

*Summary & Discussion.* In this section, we review identification-based and identification-free approaches to evaluating the self-expression capabilities of LLMs. Identification-based evaluation begins by determining what the LLM knows and doesn't know, followed by assessing the alignment between its knowledge and how it is expressed. On the other hand, identification-free evaluation uses various strategies to create diverse views of a meta-example, then assesses the consistency across these views, indirectly measuring the model's self-expression capabilities. Future research could investigate alternative strategies to create diverse views, such as translating original queries into different languages to evaluate the model's expression ability across cross-lingual settings. Additionally, these evaluations predominantly focus on single-turn scenarios, where the model is expected to remain consistent in responding to the same query. Future studies could extend to multi-turn scenarios, where the model should maintain consistency with the conversation history over time.

## 4 Improvement of Self-knowledge

Many studies aim to improve the self-knowledge capabilities of LLMs. One line of research teaches them to articulate *"I don't know"*. Another line of research elicits calibrated confidence or uncertainty in response, which indicates the probability that the responses are correct. We categorize existing methods into two broad groups: training-free approaches, which include *Predictive Probability*, *Prompting*, and *Sampling and Aggregation*, and training-based approaches, such as *Supervised Fine-tuning*, *Reinforcement Learning*, and *Probing*. An overview of these methods is provided in Fig. 5.

### 4.1 Training-free Approaches

**Predictive Probability.** A straightforward approach to providing confidence is computing predictive probability, which has been extensively explored in NLP classification tasks with masked language models (Xiao et al., 2022). In the era of LLMs, the predictive probability of an output is formalized as

$$\log p(\boldsymbol{y}|\boldsymbol{x}) = \sum_{t=1}^{T} \log(y^t|y_{<t}, \boldsymbol{x}), \tag{12}$$

where $\boldsymbol{x}$ and $\boldsymbol{y}$ represent prompt and output respectively. As this measure is biased towards output length $T$ (Wu et al., 2016), the length-normalized version is frequently used by dividing $\log p(\boldsymbol{y}|\boldsymbol{x})$ with $T$ (Adiwardana et al., 2020; Malinin & Gales, 2021; Si et al., 2023a; Kuhn et al., 2023). Kadavath et al. (2022) indicate that the predictive probability of LLM is well-calibrated on multiple-choice tasks ($T = 1$) and the calibration improves with the capability of LLM. However, empirical experiments show that predictive probability is less suitable for free-form generation tasks ($T > 1$) (Kuhn et al., 2023; Ren et al., 2023b). Inspired by this observation, Ren et al. (2023b) convert free-form generation into multiple-choice selection by sampling multiple candidate answers and forming them into a multiple-choice format, with the predicted probabilities of the options serving as the confidence. One potential reason for the weakness of predictive probability in free-form generation is that token probability captures lexical confidence instead of semantic confidence (Kuhn et al., 2023), which is more desired in applications. To better capture semantics, Duan et al. (2024) reweight the token probability with a relevance score, which represents the semantic change before and

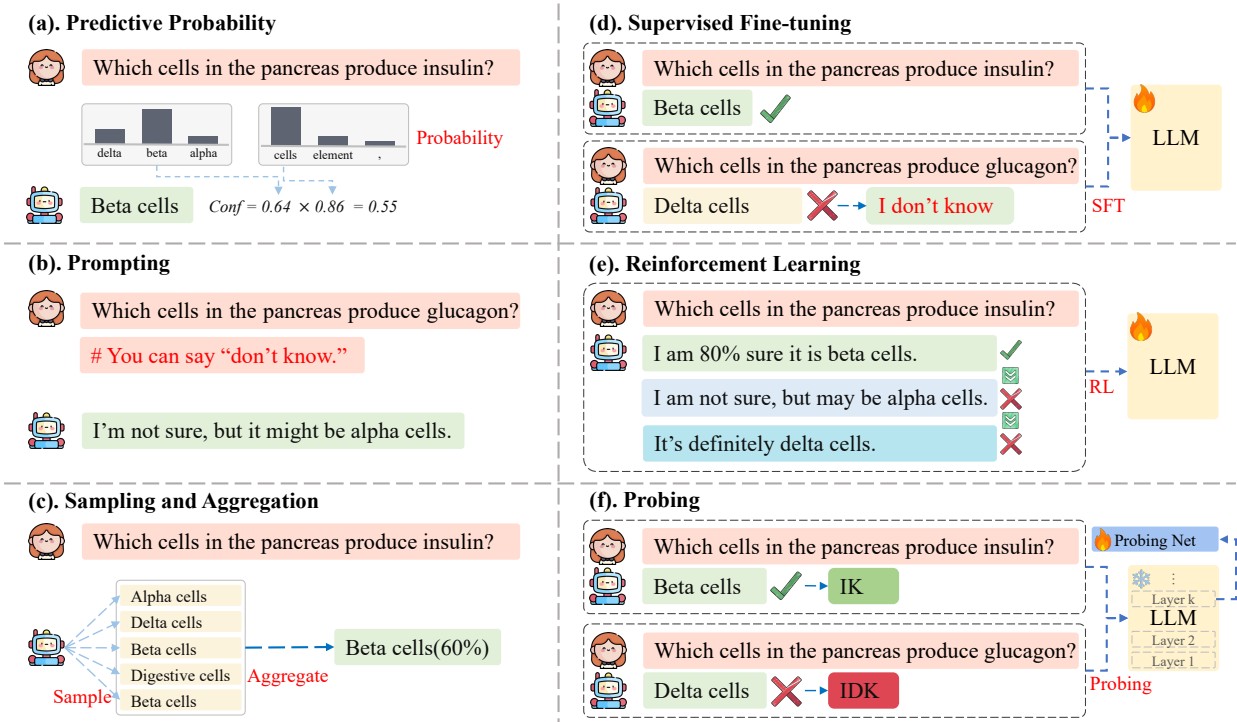

Figure 5: Improvement of self-knowledge, encompassing both training-based and training-free approaches.

after the token is removed. A fundamental limitation of predictive probability is the requirement of token-likelihood, which might be inaccessible for closed-source LLMs, such as GPT-4 (Achiam et al., 2023).

*Summary & Discussion.* Through pre-training on large corpora with language modeling loss, LLMs have demonstrated well-calibrated token-level predictions in constrained token-level tasks, such as multiple-choice selection (Kadavath et al., 2022) and the completion of prepositional verbs (Ilia & Aziz, 2024). However, sequence-level calibration remains challenging. The sequence of token probabilities reflects both aleatoric and epistemic uncertainty (Der Kiureghian & Ditlevsen, 2009). In the context of LLMs, aleatoric uncertainty refers to variations in synonyms or synthetic forms, whereas epistemic uncertainty is related to the semantic meaning, which is more directly tied to the overall performance. To more accurately capture model behavior based on predictive probability, it is crucial to minimize the effects of aleatoric uncertainty. One possible solution is to constrain outputs, such as by transforming multiple outputs into a multiple-choice task, as in Ren et al. (2023b). Additionally, tokens can be reweighted based on their semantic importance, as in (Duan et al., 2024), or outputs with similar semantic content but diverse forms can be clustered, and semantic uncertainty can be estimated based on these clusters (Kuhn et al., 2023).

**Prompting.** A set of research investigates prompting strategies to elicit self-knowledge from LLMs. We provide an overview of these strategies in Table 4. In earlier studies, Kadavath et al. (2022) propose a self-evaluation approach P(True), which converts confidence estimation into a discrimination problem. In particular, they prompt LLM to identify whether its answer is true or false given the question as a context, then the predicted probability of "true" serves as its confidence in this answer. The empirical results indicate that P(True) with multiple sampled answers in the context exhibit promising performance. Drawing from psychological and cognitive research, Zhao et al. (2024) propose a fact-and-reflection strategy. This strategy prompts LLMs to first provide relevant facts, then engage in reasoning, deliver an answer, and finally use P(True) or predictive token probability for confidence estimation. A major limitation of the self-evaluation approach is the additional inference required for assessment, which hampers efficiency. Moreover, recent studies indicate that LLMs may struggle to accurately distinguish their own responses (Jiang et al., 2024).

Table 4: Overview of prompting methods for improving self-knowledge. {Response} is LLM's response.

| Method | Prompt |
|---|---|
| P(True) (Kadavath et al., 2022) | Here are some brainstormed answers: {Response}
Proposed Answer: {Response}
Is the proposed answer: (A) True (B) False
The proposed answer is: |
| Fact-and-reflection (Zhao et al., 2024) | Provide supporting facts and the sources: {Response}
Provide the reasoning process: {Response}
Provide the final answer: {Response}
Is the proposed answer: (A) True (B) False
The proposed answer is: |
| Instruction (Yin et al., 2023) | Provide your answer. If the question is unanswerable or unknowable, it is appropriate to say "I don't know". |
| In-context Learning (Yin et al., 2023) | Q: What is the highest building in New York?
A: The highest building is the One World Trade Center.
Q: Will nuclear war break out in the world in 2050?
A: It is impossible to predict with certainty. I don't know.
Q: [...] |
| Top-K 1S (Tian et al., 2023) | Provide your K best guesses and the probability that each is correct. |
| Top-K 2S (Tian et al., 2023) | Provide your K best guesses. {Response}
Provide the probability that each is correct. |
| CoT 1S (Xiong et al., 2024) | Analyze step by step, provide your answer and your confidence in this answer. |
| CoT 2S (Tian et al., 2023) | Analyze step by step and provide your answer. {Response}
Provide the probability that the answer is correct. |
| Linguistic (Tian et al., 2023) | Provide your answer, and describe the likelihood of your answer being correct using one of the following expressions: {Almost certain, Likely, . . . , Almost no chance} |
| Self-probing (Xiong et al., 2024) | Possible answer: {Response}
How likely is the above answer to be correct? Analyze the possible answer, provide your reasoning concisely, and give your confidence in this answer. |
| Multi-step (Xiong et al., 2024) | Break down the problem into K steps, think step by step, give your confidence in each step, and then derive your final answer and your confidence in this answer. |

Another line of research prompts LLMs to verbalize self-knowledge. Yin et al. (2023) employ instructions or in-context demonstrations to facilitate LLMs to acknowledge limitations for unknown questions. Tian et al. (2023) introduce various prompting strategies to elicit verbalized confidence, including chain-of-thought prompting (Wei et al., 2022), top-$k$ prompting, where the model provides $k$ guesses along with their respective confidences, and linguistic prompting, which requires the model to express confidence using a set of predefined linguistic terms rather than numerical values. Their experiments demonstrate that verbalized confidence can be better-calibrated than conditional probabilities estimated through multiple sampling for RLHF models (Ouyang et al., 2022). Inspired by human conversations, Xiong et al. (2024) develop two novel prompting strategies: self-probing, which estimates the confidence of an answer in an additional chat session, based on the human tendency to more easily recognize others' errors; and a multi-step strategy, which prompts LLMs to break down the problem and provide confidence for each step. Despite significant advancements, the effectiveness of verbalizing self-knowledge remains contentious. Yona et al. (2024) indicate that LLMs struggle to faithfully convey uncertainty, such as hedging when they are confident and providing definitive answers despite underlying uncertainty. Xiong et al. (2024) suggest that when LLMs verbalize confidence, they are more likely to mimic human expressions of confidence rather than genuinely assess the answer

based on their knowledge. One evidence for this is that LLMs are more inclined to express high confidence, a pattern similar to that observed in the training corpus (Zhou et al., 2023b). Additionally, Krause et al. (2023) investigate the expression of uncertainty in LLMs within multilingual contexts and observe a substantial drop in performance for low-resource languages compared to English.

*Summary & Discussion.* Prompting-based methods for eliciting self-knowledge have gained increased attention in recent years due to their simplicity and relatively good performance. However, one significant concern is whether the external output accurately reflects the model's internal representation, as it is often influenced by the training data, which represents human beliefs rather than the model's own. Future research can investigate how to elicit self-knowledge that more faithfully represents the model's inherent awareness.

**Sampling and Aggregation.** Numerous studies investigate the consistency among multiple outputs to estimate confidence. Typically, they use temperature sampling to obtain diverse outputs based on the same prompt, with the temperature controlling the randomness (Zhou et al., 2022; Kuhn et al., 2023; Lyu et al., 2024). Alternatively, Xiong et al. (2024); Yang et al. (2024a) improve diversity by rephrasing the original prompt instead of sticking to a fixed one. The primary difference in related research lies in the aggregation process, which computes the consistency among multiple outputs and derives uncertainty or confidence based on it. Zhou et al. (2022) compute the answer frequency in multiple outputs as confidence. Xiong et al. (2024); Lyu et al. (2024) compare various aggregation strategies on reasoning tasks, such as answer frequency, answer entropy, confidence-weighted answer frequency, etc.

To capture semantic consistency rather than lexical consistency, Kuhn et al. (2023) propose semantic entropy. They begin by clustering outputs based on their entailment measured by a natural language inference (NLI) model (Williams et al., 2018), and then consider the entropy of these clusters as semantic entropy. Lin et al. (2024b) employ NLI to assess consistency and investigate multiple strategies to convert it into measures of uncertainty or confidence, incl2uding the number of clusters, the degree matrix, and other related metrics. Fadeeva et al. (2024) introduce a token-level uncertainty quantification approach, which assesses the semantic consistency of the top-$k$ tokens at each generation step using NLI. Beyond NLI, Manakul et al. (2023) explore alternative methods for evaluating consistency, such as BERTScore (Zhang et al., 2020b), n-gram analysis, and prompting with LLMs. Yadkori et al. (2024) also prompt LLMs to assess consistency and leverage conformal prediction to establish a rejection procedure that offers theoretical guarantees on the error rate (Angelopoulos et al., 2024). Instead of measuring consistency in the language space, Chen et al. (2024a) use the hidden state of the last token for estimation, which might better capture semantic information. They construct a covariance matrix of the hidden states from multiple outputs and use its logarithm determinant as a measure of consistency, representing the differential entropy in the embedding space.

For long-form generations, Huang et al. (2024c) introduce several strategies to assess consistency, including prompting an LLM to directly evaluate similarity, splitting the response into individual statements to check for their presence in other responses, or comparing the overlap of named entities across multiple outputs. Similarly, Farquhar et al. (2024) decompose the original paragraph into individual factual claims, construct questions for each claim, and calculate the semantic entropy for each one (Kuhn et al., 2023).

*Summary & Discussion.* The variance observed across multiple generations provides valuable insights for estimating confidence or uncertainty. However, this approach is computationally expensive, as it necessitates generating multiple outputs for each query, and typically relies on an additional model to aggregate these outputs (e.g., NLI model). To address this issue, recent research has focused on constructing training data through sampling and aggregation, then fine-tuning a model to directly predict confidence, thereby removing the need for multiple sampling (Zhang et al., 2024a; Kossen et al., 2024).

### 4.2 Training-based Approaches

Training-free approaches, particularly those based on predictive probability or prompting strategies, show promise but have inherent limitations. Although pre-training corpora include uncertain expressions (Zhou et al., 2023b), these are aligned with human capabilities rather than those of LLMs. As a result, LLMs tend to mimic human uncertainty instead of faithfully express their own confidence levels (Zhou et al., 2023b;

Xiong et al., 2024). To address this issue, training strategies can be applied to better align LLMs' expression of self-knowledge with their actual capabilities.

**Supervised Fine-tuning.** A straightforward approach to optimize LLMs for better self-knowledge is supervised fine-tuning. One line of research fine-tunes LLMs to verbalize *"I don't know"* when they lack relevant knowledge. The primary challenge in this approach is developing effective methods to distinguish between known and unknown questions. Yang et al. (2024b); Zhang et al. (2024a); Cheng et al. (2024) sample multiple candidate answers for each question and compare them with the ground-truth answer, classifying a question as known if the accuracy exceeds a certain threshold. In contrast, Chen et al. (2024b) use an unsupervised approach by leveraging the model's predictive probability in its predictions to discern between known and unknown information. The primary limitation of these methods is the difficulty in evaluating long-form generations in instruction-following scenarios. To address this problem, Wan et al. (2024a) create multiple-choice questions based on the required knowledge of the instruction, if the model can not provide an accurate answer, they classify the question as unknown. Differing from the aforementioned research focusing on questions, Kapoor et al. (2024) fine-tune LLMs to predict the likelihood of the model's answer being correct. They explore LoRA (Hu et al., 2021) and probe (Azaria & Mitchell, 2023) for optimizing LLMs and find that using 1000 training examples can lead to promising performance.

In addition to distinguishing between known and unknown knowledge, another area of research focuses on fine-tuning LLMs to provide confidence estimates for their responses. Lin et al. (2022a) fine-tune GPT-3 to verbalize confidence for arithmetic questions, with the target confidence corresponding to the empirical precision of GPT-3 on that type of question. Similarly, Yang et al. (2024b); Han et al. (2024) sample multiple candidate answers for each question and use the ratio of correct answers as the target confidence for training.

To accommodate the black-box setting, several works have also fine-tuned additional models for confidence prediction. Ulmer et al. (2024) collect LLM-generated answers to a set of questions, cluster these question-answer pairs based on sentence similarity, and evaluate the accuracy of the LLM within each cluster. They then use this cluster-level accuracy as the confidence label for each question-answer pair. Subsequently, they fine-tune a DeBERTa model (He et al., 2023) to predict confidence based on these pairs. In another approach, Liu et al. (2024d) employ a white-box model to capture the hidden states of predicted answers generated by a target model, such as GPT-4. They then train an additional model to predict confidence based on these hidden states, potentially capturing more semantic information than the raw text. Additionally, Fathullah et al. (2024) fine-tune an encoder model conditioned solely on the input source to predict uncertainties prior to generation, which leads to a significant improvement in efficiency.

*Summary & Discussion.* Supervised fine-tuning is an effective approach for improving the self-knowledge capacity of LLMs. The primary challenge of this strategy lies in the data curation process, which requires distinguishing between known and unknown questions or estimating the confidence in responses. Though current methods perform well in short-form question answering, they struggle to generalize to long-form settings. Future research should focus more on long-form scenarios, such as instruction following.

**Reinforcement Learning.** Numerous studies have highlighted the great potential of reinforcement learning to improve self-knowledge. Cheng et al. (2024); Xu et al. (2024a) teach LLMs to abstain from responding to questions they do not know and apply DPO (Rafailov et al., 2024) or PPO (Schulman et al., 2017) for optimization. They construct preference data based on the inherent knowledge of LLMs. Specifically, if the model correctly answers a question, the preferred response is the correct answer, and the rejected response is *"I don't know"*. Conversely, if the model answer incorrectly, the preferred response is *"I don't know"*, while the rejected response is the incorrect answer. More simply, Gao et al. (2024) create preference pairs by using LLMs to judge both honesty and helpfulness, then use DPO for optimization to address the potential conflict between honesty and helpfulness (Liu et al., 2024f). To provide more fine-grained information, Xu et al. (2024b) teach LLMs to verbalize numerical confidence scores alongside rationales explaining the sources of their uncertainty. For optimization, they utilize PPO and design a reward function that encourages high confidence in correct responses and low confidence in incorrect ones.

Recent studies explicitly model the human-AI interaction process by simulating a *"listener"* using an LLM, who makes decisions based on the response from a *"speaker"* LLM. They fine-tune the speaker to either

refuse to answer unknown questions or express well-calibrated confidence in its response, so that the listener could make proper decisions accordingly, such as accepting or rejecting the response. Specifically, Stengel-Eskin et al. (2024) train LLMs to articulate appropriate implicit (e.g., hedges) or explicit confidence markers (e.g., numeric confidence) using DPO. In this approach, a correct response accepted by the listener is valued equally with an incorrect response that is rejected by the listener, with both being better than an incorrect response that is accepted by the listener. Additionally, Band et al. (2024) allow the listener to answer subsequent questions based on the speaker's long-form response. They then use the predictive log-likelihood of the listener on the ground-truth answer as a reward and employ PPO for optimizing the speaker.

*Summary & Discussion.* Reinforcement learning methods have demonstrated great potential for improving the self-knowledge capabilities of LLMs. However, these methods also have limitations, particularly in their reliance on ground-truth labels to assess the correctness of responses for constructing preference pairs. Additionally, current PPO-based strategies provide rewards based on the ground-truth labels (Xu et al., 2024b; Band et al., 2024), thereby limiting the exploration space during training. Future research could focus on developing unsupervised methods to provide supervision for reinforcement learning, such as *Predictive Probability*, *Prompting* and *Sampling and Aggregation*.

**Probing.** Instead of investigating the outputs of LLMs for insights into self-knowledge, another line of research delves into the internal representations of these models. Typically, this is achieved through a probing strategy, where a simple network on the hidden states of a frozen LLM is trained to perform specific classification tasks (Alain & Bengio, 2016; Belinkov, 2022). In an earlier study, Kadavath et al. (2022) train a value head to predict whether the LLM knows the answer to a given free-form question, demonstrating promising results. Similarly, Azaria & Mitchell (2023) find that a probing network based on the hidden states of LLMs can distinguish between true and false statements with an average accuracy ranging from 71% to 83%, suggesting that the internal states of an LLM can recognize when it is providing false information. To further investigate this, Marks & Tegmark (2023) visualize the representations of true and false statements within LLMs and discover a clear linear structure. Moreover, Ji et al. (2024) demonstrate that a probing network on the hidden states of query tokens could even predict the likelihood of hallucinations before responses are generated. Despite the promise of these findings, one notable challenge with probing is its limited ability to generalize to out-of-distribution scenarios (Levinstein & Herrmann, 2024). To address this, Liu et al. (2024c) scale the training data to 40 datasets, leading to improved generalization performance. Liu et al. (2023) demonstrate that probing consistently outperforms the prompting method P(True) (§4.1) in evaluating prediction correctness, with its superiority mainly attributed to better calibration on uncertain predictions. Moreover, performance can be further improved by ensembling the two methods.

The probing network can also be developed without supervision. Burns et al. (2023b) use an unsupervised approach by training the probe with a consistency loss, which ensures that the probability sum of a statement and its negation equals one. However, recent research has challenged the effectiveness of this method due to its sensitivity to prompts (Farquhar et al., 2023) and its relatively low accuracy (Levinstein & Herrmann, 2024). In contrast, Kossen et al. (2024) train the probing network to predict the semantic entropy of each example (Kuhn et al., 2023), which has shown promising performance in hallucination detection and has demonstrated better generalization compared to probes trained to assess whether a statement is true or false.

*Summary & Discussion.* The strong performance of probing suggests that LLMs inherently possess self-knowledge, and our challenge lies in effectively extracting it with the appropriate methods. Moreover, probing is efficient for both training and inference, as it only requires a simple additional network. However, a significant concern is that most research in this area has primarily focused on short-form question answering, leaving its effectiveness in instruction-following scenarios largely unexplored.

## 5 Improvement of Self-expression

Numerous studies aim to improve the ability of LLMs to faithfully express their knowledge in responses. To provide a clearer understanding of current research, we classify the approaches into two main categories: training-free approaches, which include *Prompting, decoding-time intervention, sampling and ag-*

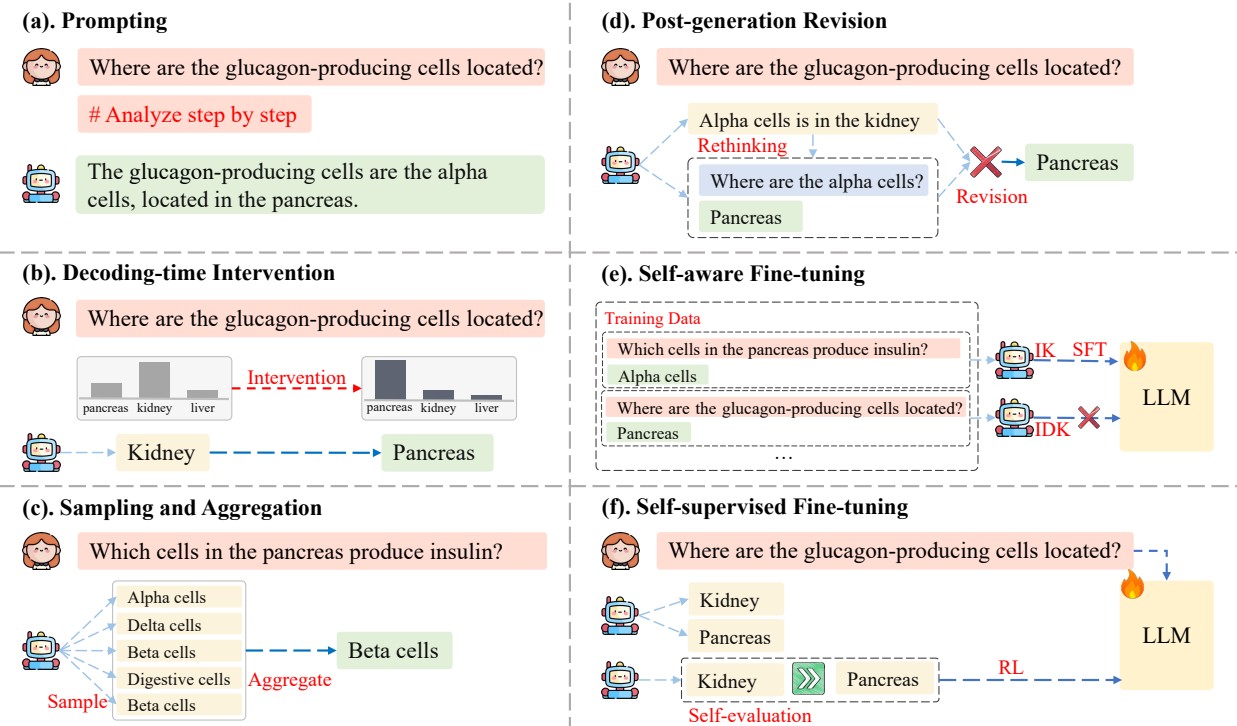

Figure 6: Improvement of self-expression, encompassing both training-based and training-free approaches.

*gregation*, and *post-generation revision*, and training-based approaches including *self-aware fine-tuning* and *Self-supervised fine-tuning*. Take an overview of these methods in Fig. 6.

## 5.1 Training-free Approaches

**Prompting.** Pre-training enables LLMs to retain factual knowledge, but it is less effective at teaching them how to compose individual facts to answer complex questions (Press et al., 2023), leading to challenges in fully express their internal knowledge. One promising approach to address this issue is by using well-designed prompting strategies, as summarized in Table 5. Chain-of-thought prompting (CoT) (Wei et al., 2022) encourages LLMs to engage in a step-by-step reasoning process by providing few-shot demonstrations. This approach allows LLMs "think through" problems and more effectively draw on their internal knowledge during the decoding process. Later research explores zero-shot prompting by simply adding the phrase *Let's think step by step* to the prompt (Kojima et al., 2022) to encourage step-by-step thinking.

Further studies focuses on structuring the generation process more explicitly. Zhou et al. (2023a) introduce least-to-most prompting, where the model first breaks a complex problem into smaller sub-questions, solves them one by one, and then combines the answers to tackle the original problem. Similarly, Self-ask (Press et al., 2023) prompts the LLM to decide when follow-up questions are needed and, if so, generates both the question and answer iteratively. Drawing from the cognitive skill of abstraction (Lachmy et al., 2022), step-back prompting (Zheng et al., 2024) allows LLMs to identify high-level abstractions, such as key concepts or principles, and use them to guide the generation process. In zero-shot settings, Wang et al. (2023c) propose plan-and-solve prompting, where the model first outlines a plan and then executes it step-by-step. Zhao et al. (2024) introduce fact-and-reflection, where the model first recalls relevant knowledge and then reflects on it to arrive at the final answer.

*Summary & Discussion.* CoT prompting encourages LLMs to express their internal knowledge through step-by-step generation. However, its success depends largely on well-crafted prompts, which may be suboptimal and lack clear explainability. Moreover, current LLMs are often sensitive to prompt variations, raising

Table 5: Overview of prompting methods for improving self-expression. The content within {} represents the response of LLMs.

| Method | Prompt |
| --- | --- |
| Chain-of-Thought (Wei et al., 2022) | Q: Where is the highest mountain in the world?
A: The highest mountain is Mount Everest, and it is located on the border of Nepal and Tibet (China). |
| Zero-shot CoT (Kojima et al., 2022) | Let's think step by step. |
| Least-to-Most (Zhou et al., 2023a) | Q: Where is the highest mountain in the world?
Sub-Q1: {Which is the highest mountain in the world?}
Sub-A1: {Mount Everest}
Sub-Q2: {Where is Mount Everest?}
Sub-A2: {border of Nepal and Tibet (China)} |
| Self-ask (Press et al., 2023) | Q: Where is the highest mountain in the world?
Are follow up questions needed: {Yes}
Q1: {Which is the highest mountain in the world?}
A1: {Mount Everest}
Are follow up questions needed: {Yes}
Q2: {Where is Mount Everest?}
A2: {Border of Nepal and Tibet (China)} |
| Step-back (Press et al., 2023) | Q: Which school did Kaiming He attend in November 2010?
Stepback Q: {What was Kaiming He's education history?}
Stepback A: {B.S. in THU, 2007; Ph.D. in CUHK, 2011}
Final A: {The Chinese University of Hong Kong} |
| Plan-and-solve (Wang et al., 2023c) | Let's first understand the problem and devise a plan to solve the problem. Then let's carry out the plan and solve the problem step-by-step. |
| Fact-and-reflection (Zhao et al., 2024) | Q: Where is the highest mountain in the world?
What are the facts needed to answer this question?
Facts: {1.The highest mountain is Mount Everest. 2.Mount Everest is located on the border of Nepal and Tibet (China).}
What is your reasoning?
Reasoning: {The highest mountain, Mount Everest, is located on the border of Nepal and Tibet (China).}
Final Answer: {Border of Nepal and Tibet (China)} |

concerns about the generalization capabilities of this method. Future research could explore the underlying mechanisms behind the effectiveness of CoT prompting or develop methods for automatically creating prompts that are both optimal and robust.

**Decoding-time Intervention.** Numerous studies concentrate on eliciting the internal knowledge of LLMs through the decoding process. Li et al. (2024a) reveal a substantial gap between generation accuracy and probing accuracy as measured by a classifier on the hidden states of LLMs, suggesting that LLMs "know" more than they "say". To fully leverage this potential, they modify the activations of LLMs using truthful directions derived from the hidden states of true and false statements. Chen et al. (2024c) discover a strong correlation between the probability of hallucinations and contextual activations, which represents the mapping between the hidden states of context tokens and the predictive token. Using this information, they steer the decoding process to effectively reduce the occurrence of hallucinations.

Instead of investigating the hidden states, many studies directly modify the predictive distribution. Notably, contrastive decoding is extensively explored (Li et al., 2023a), where the logit difference between an expert model and an amateur model is utilized to steer generation, which amplifies the advantages of the expert and reduce the disadvantages of the amateur. Specifically, Chuang et al. (2024) employ the predictive distribution from the higher layer as the expert and that from the lower layer as the amateur, aiming to emphasize the

factual knowledge embedded in the higher layer. Similarly, Zhang et al. (2023a) utilize the original LLM as the expert and a hallucination-prone LLM as the amateur to amplify the knowledge within the original model and reduce its tendency to fabricate information. Besides, Shi et al. (2024a) adopt the strong and weak activations of an MoE model as the expert model and the amateur model respectively, aiming to emphasize the reasoning capability of the MoE model's strong activations. In addition to focusing on parametric knowledge, Shi et al. (2024b) propose context-aware decoding, where both the expert and the amateur share the same LLM, but only the expert has access to context, thereby highlighting the importance of contextual knowledge in responses. The idea of contrastive decoding is also applicable in multimodal scenarios. Leng et al. (2024b) use a model with original visual inputs as the expert and the same model with distorted visual inputs as the amateur, with the goal of highlighting the role of visual inputs in shaping responses.

*Summary & Discussion.* Decoding-time intervention shows great promise in unlocking the potential of LLMs, but current research faces two primary challenges. First, the generalization is limited, as most methods have been concentrated on specific domains, such as factuality or reasoning, leaving their effectiveness in general instruction-following scenarios relatively underexplored. Additionally, these approaches may incur extra computational costs, such as an extra forward pass in contrastive decoding. Future research could develop strategies to address these challenges.

**Sampling and Aggregation.** A straightforward approach for eliciting faithful knowledge from LLMs is through sampling and aggregation. This involves sampling multiple outputs and then aggregating them to derive the most consistent one, which is expected to more accurately reflect the model's knowledge. Wang et al. (2023d) sample a set of reasoning paths for a single query and then aggregate their answers by majority voting. This simple strategy achieves great performance in various reasoning tasks. Instead of aggregating answers, Chen et al. (2023b) prompt an LLM to aggregate multiple free-form responses based on majority consensus for open-ended generation. To make the most of multiple outputs, Thirukovalluru et al. (2024) split each output into several atomic parts, cluster them using sentence embeddings, remove clusters with fewer elements, and summarize the remaining clusters to produce a final consistent output. Similarly, Wang et al. (2024c) propose prompting an LLM to integrate and derive the final output based on the majority consensus from multiple outputs.

*Summary & Discussion.* Intuitively, consistent content across multiple generations tends to faithfully reflect the model's knowledge. While this approach demonstrates strong performance, the multiple sampling process incurs substantial computational costs, limiting its applicability in real-world settings. To overcome this challenge, future research could focus on constructing training data through sampling and aggregation, then fine-tuning the model with this data to internalize this capability.

**Post-generation Revision.** Another approach involves post-generation refinement, where the response is modified to reduce inconsistencies with the model's knowledge. Dhuliawala et al. (2023a) first prompt the LLM to generate a list of questions designed to verify the atomic facts in its initial response, and then provide answers to these questions individually. Following this, the LLM is prompted to check the consistency between the initial response and the answers, making any necessary revisions. Similarly, Varshney et al. (2023) identify key concepts from the response, such as entities and keywords, evaluate their confidence using token probability, and retrieve external knowledge to validate and revise low-confidence concepts, which are more likely to be fabricated. Analogously, Zhao et al. (2023) assess the consistency across multiple responses as described by Wang et al. (2023d), then revise the less consistent ones using external knowledge.

*Summary & Discussion.* Post-generation revision offers an additional chance to correct inaccurate knowledge expression. The primary concern, however, is the increased computational cost. A possible direction for future research is to create training data based on this strategy and then fine-tune the LLM accordingly. For example, the original outputs and their revised versions could be treated as preference pairs, allowing methods like DPO or PPO to align the LLM with the desired attributes of the revised outputs.

## 5.2 Training-based Approaches

**Self-aware Fine-tuning.** Recent research suggests that fine-tuning LLMs with new knowledge may diminish their ability to express knowledge accurately, as this process can teach them to fabricate information

beyond their internal knowledge (Gudibande et al., 2023; Lin et al., 2024a; Gekhman et al., 2024). To address this issue, many studies have started taking the knowledge boundaries of LLMs into account during fine-tuning, an approach we refer to as self-aware fine-tuning. Yang et al. (2024b); Zhang et al. (2024a); Cheng et al. (2024) fine-tune LLMs to explicitly state *"I don't know"* when they lack adequate knowledge, thereby reducing the risk of generating fabricated information. Alternatively, Wan et al. (2024a) propose discard tuning, where examples are discarded when the model lacks the necessary knowledge, and open-book tuning, which incorporates reference knowledge into the context during fine-tuning to prevent the models from learning to fabricate content. More details on the knowledge identification methods used in these studies can be found in §4.2. Kang et al. (2024) introduce a conservative reward model that encourages less detailed responses in situations where the LLM is unfamiliar with the queries.

*Summary & Discussion.* Self-aware fine-tuning has demonstrated great potential in alleviating the tendency to fabricate information. The primary challenge lies in distinguishing between what the model knows and doesn't know. For future research, RL-based self-ware fine-tuning could be a valuable area for exploration, as it allows the LLM to explore broader spaces during training, potentially providing a more accurate reflection of its knowledge boundary.

**Self-supervised Fine-tuning.** Another line of research fine-tunes LLMs to improve the expression of knowledge by leveraging supervision from their internal knowledge. Tian et al. (2024) initiate this process by prompting GPT-3.5 to extract multiple atomic claims from long-form generations. For each claim, they create a verification question, sample multiple answers from the LLM, and then calculate the consistency score across these answers. Based on the average consistency score of all claims, they construct preference pairs and employ DPO for optimization. Similarly, Zhang et al. (2024c) propose a method that use self-evaluation to verify each claim and also utilizes DPO for optimization. Simply, Lin et al. (2024a) prompt the LLM to generate responses for fact-based instructions, which are then used to fine-tune the model. After the fine-tuning phase, they create preference pairs by sampling responses from the LLM and use the model itself to assess these responses. Finally, they apply DPO for further refinement.

*Summary & Discussion.* Self-supervised fine-tuning has proven effective in improving LLMs' knowledge expression ability. However, a major concern is the quality of self-provided supervision signals, as these models may produce incorrect statements, and recent studies (Huang et al., 2024a; Jiang et al., 2024) have raised doubts about their self-evaluation abilities. Future research could aim to more thoroughly investigate the reliability of self-supervised fine-tuning.

**Others.** In addition to the previously mentioned methods, there are other techniques aimed at improving the ability of LLMs to express their knowledge. Wei et al. (2023) construct simple synthetic data to train LLMs to avoid sycophancy, they reformat existing publicly available NLP datasets by adding user opinions that are independent of the correctness of the final answer and then fine-tune the models using this data. To achieve consistent model outputs across various prompts, Zhou et al. (2022); Cao et al. (2024) apply a consistency loss, which regularizes the outputs of semantically equivalent prompts to remain the same.

# 6 Future Work

In this section, we discuss several unresolved research challenges associated with honesty and provide insights into potential research avenues.

**Objective or Subjective.** A central debate in current research on the honesty of LLMs revolves around whether honesty should be considered a subjective or objective concept. Askell et al. (2021); Kadavath et al. (2022) describe honesty as the ability to provide accurate information along with calibrated confidence reflecting the correctness of its answers. In contrast, Evans et al. (2021); Lin et al. (2022a) view honesty as the model's ability to express its own beliefs. The former takes an objective approach, aligning with world knowledge, while the latter adopts a subjective perspective, focusing on the model's internal state. The objective perspective better suits human needs, as people generally value accurate and truthful information. However, this approach presents challenges for optimizing models because there is often a gap between the model's knowledge and world knowledge, necessitating additional supervision during training to *distinguish*

*between true and false information.* This becomes particularly challenging for future superhuman models, where human oversight may be limited to providing only weak supervision (Burns et al., 2023a). Conversely, the subjective perspective might require less supervision, as it primarily focuses on the model's ability to express its own knowledge, but the challenge lies in *differentiating between known and unknown knowledge.* Nonetheless, even if a model can fully articulate its knowledge, problems arise when that knowledge is incorrect. Both the objective and subjective perspectives have distinct advantages and challenges, and resolving this debate is crucial for further research progress.

**Knowledge Identification.** As stated in previous sections, knowledge identification has been the primary challenge in both evaluation and methodological approaches. However, the exact definitions of what should be considered known or unknown remain unclear. Existing research typically follows two mainstream strategies: a supervised approach, which distinguishes them based on the correctness of responses, and an unsupervised approach, which differentiates them based on the uncertainty in responses. Both strategies depend on the external expression of LLMs, but what if these models struggle to express what they know? Indeed, several studies have highlighted the discrepancy between LLMs' internal knowledge and what they express (Liu et al., 2023; Li et al., 2024a). Ignoring this issue could limit the potential of LLMs to fully express their knowledge. Therefore, in addition to focusing on external expression, future research could explore techniques that utilize the inherent knowledge LLMs possess to differentiate between known and unknown information (Burns et al., 2023b; Hernandez et al., 2023; Wang et al., 2024b). For instance, researchers could aim to recover the knowledge embedded in model parameters or present in the contexts even in cases when LLMs fail to express it faithfully in their outputs (Burns et al., 2023b).

**Honesty in Instruction-following.** Current research on honesty focuses primarily on knowledge-intensive question answering, particularly those involving short-form answers, while largely overlooking instruction-following scenarios that are more desired in real-world applications. Instruction-following differs from question answering as it requires broader capabilities and typically involves long-form generation. To address this gap, future research can establish evaluation methods and benchmarks to assess the honesty of LLMs in instruction-following. Additionally, they can explore methods for improvement, such as prompting, supervised fine-tuning, and reinforcement learning.

**Honesty on In-context Knowledge.** As noted in §2.1, LLMs possess two types of knowledge: internal parametric knowledge acquired through training and external in-context knowledge. While most existing research emphasizes the honesty of parametric knowledge, the honesty of in-context knowledge has received less attention. However, in real-world applications of LLMs, in-context knowledge also plays a vital role in generation, particularly in retrieval-augmented and long-context scenarios. Therefore, we advocate for future research to devote more attention to the honesty of in-context knowledge.

**Honesty in Various Models.** Research on honesty has primarily focused on transformer decoder-based LLMs. However, other popular models also deserve attention, including multimodal LLMs such as GPT-4V (Achiam et al., 2023) and Gemini (Team et al., 2023), models with novel architectures such as Mamba (Gu & Dao, 2023), and compressed models using compression techniques such as quantization and pruning (Wan et al., 2024b). We believe that these models also merit further exploration in future research.

## 7 Discussion & Conclusion

**Discussion.** Through a comprehensive review of existing research on the honesty of LLMs, we broadly categorize the honesty of LLMs into two key capabilities: self-knowledge and self-expression. Self-knowledge emphasizes the discriminative ability of LLMs to recognize and acknowledge the boundaries of their own knowledge, thereby mitigating undesirable behaviors such as hallucinations and overconfidence. In contrast, self-expression highlights the generative ability of LLMs to faithfully convey their inherent knowledge, whether derived from internal parameters or external context. Despite significant progress in this area, several key challenges remain that warrant further attention:

- Research on self-knowledge has primarily focused on short-form question-answering scenarios. However, it remains unclear how to enable an LLM to express its self-knowledge in real-world, long-form settings, particularly those that are complex and require a thoughtful reasoning process to identify its limitations.
- Both self-knowledge and self-expression necessitate the identification of the model's inherent knowledge. Current approaches largely rely on intuitive methods, which may be suboptimal, such as determining knowledge based on its inclusion in the pre-training corpus or the model's ability to provide correct answers. More explainable identification methods, such as those based on hidden activations, are needed to ensure that the model's inherent knowledge is neither underestimated nor overestimated.
- Current research on the honesty of LLMs mainly addresses textual inputs, while multimodal scenarios, including inputs with figures or videos, remain underexplored. It is unclear whether existing methods and findings can be effectively generalized to multimodal settings.

**Conclusion.** Honesty is a crucial factor in the development of LLMs, yet current models still exhibit significant dishonest behaviors. To address these issues, this paper offers a thorough overview of research on the honesty of LLMs, including its clarification, evaluation approaches, and improvement strategies. Furthermore, we propose several potential directions for future research. We hope this survey serves as a valuable resource for researchers studying LLM honesty and encourages further exploration in this field.

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
