# OpenReview forum: "A Survey on the Honesty of Large Language Models"
_TMLR — Accepted by TMLR_

### Review · Reviewer_58Cq · 2024-11-16

**Summary Of Contributions:**

LLMs frequently express incorrect statements. This is sometimes done with poorly calibrated confidence or in apparent disagreement with what the model thinks according to probes. This paper surveys work around these challenges of "honesty" in LLMs. It divides the survey into discussions of defining honesty, evaluating it, improving it, and future work.

**Audience:**

Yes

**Claims And Evidence:**

Yes

**Requested Changes:**

See above

**Strengths And Weaknesses:**

S1: Honesty is really difficult to define. When I first saw the paper, I was skeptical that it would handle the notion of honesty well. But I think it does a good job of covering different notions. I believe this is helpful in and of itself as a contribution. I don't think I have seen any comparably good breakdowns of honesty in LLMs, and I have seen quite a few less than perfectly founded assertions that LLMs are being dishonest.

S2: I think that the survey volume is pretty impressive. I checked for a few key papers that I know of and wanted to make sure were included. Most were already were. But consider citing https://arxiv.org/abs/2402.07282 and https://arxiv.org/abs/2312.01350

S3: Good figs.

W1: I think that section 2 is missing something. There are a lot of papers, including ones discussed here, that try to study honesty in LLMs by contrasting their outputs with what knowledge probes on their internal activations say. I see at least one of these papers mentioned in 2.2 (Liu et al., 2023). But this seems unlike the others. There are also a number of other papers cited in the paper like Burns et al. (2022) study LLMs under a similar honesty paradigm.

Overall, I think this paper will be of clear value, and I think it should be accepted with minor revisions.

---

> ### Author Response · Authors · 2025-01-22
> **Response to Reviewer 58Cq (Part 1)**
>
> We sincerely appreciate your thoughtful and constructive feedback, as well as your recognition of our work in defining and systematically exploring the concept of honesty in LLMs, along with the breadth of the survey. We are grateful for your valuable insights and would like to address your concerns as follows:
>
> > **Requested Change 1: S2: I think that the survey volume is pretty impressive. I checked for a few key papers that I know of and wanted to make sure were included. Most were already were. But consider citing https://arxiv.org/abs/2402.07282 and https://arxiv.org/abs/2312.01350**
>
> Thank you for recognizing the volume and comprehensiveness of our survey. We greatly appreciate your thoughtful suggestions regarding additional references that could strengthen our work.
>
> In response to your recommendation, we have included the first paper (arXiv:2402.07282) in the **“Reinforcement Learning” part of Section 4.2** to discuss the trade-off between honesty and helpfulness, along with the second paper (arXiv:2312.01350) in **first paragraph of** **Section 2.1** to discuss the definition honesty under the context of AI Deception. All changes are highlighted in blue.
>
> ---
>
> > **Requested Change 2: W1: I think that section 2 is missing something. There are a lot of papers, including ones discussed here, that try to study honesty in LLMs by contrasting their outputs with what knowledge probes on their internal activations say. I see at least one of these papers mentioned in 2.2 (Liu et al., 2023). But this seems unlike the others. There are also a number of other papers cited in the paper like Burns et al. (2022) study LLMs under a similar honesty paradigm.**
>
> We sincerely appreciate your valuable suggestion. You are right that discussing (Liu et al., 2023) in Section 2.2 was not entirely appropriate, as this section primarily focuses on works related to definitions, whereas (Liu et al., 2023) falls under methodological analysis. We have now moved this work to **Section 4.2, in the part on probing**, and highlighted the revision in blue. You can find the updated content in the revised paper. In this part, we discuss various methods that leverage probing and internal hidden states to improve a model’s self-knowledge, including the approach mentioned in Burns et al. (2022).

---

### Review · Reviewer_dYka · 2025-01-02

**Summary Of Contributions:**

The authors provide a well-structured, well-written survey of an interesting, pertinent, and relatively new topic related to honesty in large language models.

**Audience:**

Yes

**Broader Impact Concerns:**

It would be valuable to discuss the broader impacts of honesty in LLMs more as the paper deals with a topic directly related to human interaction.

**Claims And Evidence:**

Yes

**Requested Changes:**

Request to address the points noted in the Weaknesses above.

Request to change the wording in the following phrases:
* "speak with a confident tone" - personifying LLMs can be a slippery slope
* "research...faces challenges" - the research itself does not face challenges
* "honesty is specific to each model" - ambiguous
* "after notification" - unclear

**Strengths And Weaknesses:**

Strengths:
* Well-written, well-structured, and well-organized - important features of a survey paper
* Provides a strong and un-ambiguous definition of honesty in large language models
* Excellent figures (Figure 1, Figure 2) that summarize key aspects of the paper
* Includes valuable categories of related work to address honesty in LLMs from multiple angles

Weaknesses:
* The paper does not address the scope of models being considered. For instance, is there a reason it is restricted to LLMs, and if so, are these specifically decoder-based text generation models? If not, can some exposition be included to discuss how the work may apply to other types of models?
* Other than the training-based approach sections (4.2, 5.2), the paper does not address why we should or shouldn't expect an LLM to naturally be able to express self-knowledge accurately, as in how this may interact with an LLM's pre-training objective. This is an extremely important aspect.
* The paper does not sufficiently address the implications of this question in terms of human interaction and how honesty is especially relevant and the consequences of not being honest. Generally, the paper does not provide very strong motivation for why honesty is an important issue (though of course it is, so this should be remedied easily).

---

> ### Author Response · Authors · 2025-01-22
> **Response to Reviewer dYka (Part 1)**
>
> We sincerely appreciate your detailed review and recognition of our work's merits. Below are our point-by-point responses to your concerns:
>
> > **Requested Change 1: The paper does not address the scope of models being considered. For instance, is there a reason it is restricted to LLMs, and if so, are these specifically decoder-based text generation models? If not, can some exposition be included to discuss how the work may apply to other types of models?**
>
> Thank you for your suggestion regarding the scope of models covered in our survey. In the updated version of the paper, we have discussed this point and made the following additions in the **“Honesty in Various Models” part of Section 6**.
>
> Currently, research on honesty primarily focuses on transformer decoder-based LLMs due to their prominent performance and their widespread adoption. However, many other architectures (such as Mamba) or multimodal large language models (e.g., GPT-4V) have also demonstrated great performance, and ensuring honesty in interactions with them is also necessary.
>
> Therefore, applying the insights provided in this survey and identifying the unique characteristics of these models is worthwhile. Take the self-expression capability of multimodal large language models as an example, when both the input and output are text, the content surveyed in this paper can serve as a reference. However, when the input consists of images and text,  future work will need to additionally consider the consistency between the output and the visual information.
>
> ---
>
> > **Requested Change 2: Other than the training-based approach sections (4.2, 5.2), the paper does not address why we should or shouldn't expect an LLM to naturally be able to express self-knowledge accurately, as in how this may interact with an LLM's pre-training objective. This is an extremely important aspect.**
>
> Thank you very much for this question. It is highly important as it directly impacts the validity of training-based approaches. In the updated version of the paper, we have provided an explanation at the **beginning of Section 4.2**, with the revisions highlighted in blue. You can refer to the updated paper for details.
>
> ---
>
> > **Requested Change 3: The paper does not sufficiently address the implications of this question in terms of human interaction and how honesty is especially relevant and the consequences of not being honest. Generally, the paper does not provide very strong motivation for why honesty is an important issue (though of course it is, so this should be remedied easily).**
>
> We truly appreciate this question, as it is crucial for the study of honesty. We have carefully revised the relevant content, including discussions on the significance of honesty in human society and its importance in human-AI interaction. You can find the revised content at the **beginning of Section 2** in the updated paper, highlighted in blue.
>
> ---
>
> > **Requested Change 4: Request to change the wording in the following phrases.**
>
> Thank you for your suggestions regarding the phrases in our paper. In the updated version, we have revised these expressions accordingly, with changes highlighted in blue:
>
> | No. | Comment                                                                 | Response                                                                                       |
> |-----|-------------------------------------------------------------------------|-----------------------------------------------------------------------------------------------|
> | 1   | "speak with a confident tone" - personifying LLMs can be a slippery slope | We have revised this in the second paragraph of Section 1.                                     |
> | 2   | "research...faces challenges" - the research itself does not face challenges | We have made changes in the Abstract and the third paragraph of Section 1.                     |
> | 3   | "honesty is specific to each model" - ambiguous                          | We have revised this in the third paragraph of Section 1.                                      |
> | 4   | "after notification" - unclear                                           | We have modified this in the final paragraph of Section 1.                                     |

---

### Review · Reviewer_eGs9 · 2025-01-10

**Summary Of Contributions:**

The authors present a survey about honesty in large language models. They define honesty as both being aware of its own capabilities (self-knowledge) and the ability to faithfully express its knowledge (self-expression). After defining these concepts, the paper surveys ways in which self-knowledge and self-expression are evaluated and can be improved, before giving it outlook to future work.

**Audience:**

Yes

**Broader Impact Concerns:**

I don't foresee any ethical implications of the paper due its nature as a survey.

**Claims And Evidence:**

Yes

**Requested Changes:**

* Better definitions of central concepts
   * I like that the paper surveys the field from the perspective of honesty. However, I find honesty and the notions of self-knowledge and self-expression only superficially defined. Honesty for instance is only given as a dictionary definition. But how is honest interaction defined betweem humans? How does human interaction benefit from honesty? How and when does it suffer from dishonesty? Which of these insights are applicable to human-AI interactions? How does it relate to trust? Especially when thinking of the last two questions, I think of the works of [1, 2], but especially with the rest I would like to learn more about the definition and insights from fields such as e.g. philosophy, psychology, and game theory. Not only do I think it would set the survey apart and would be very educational, but also set the survey on a solid foundation.
    * This critique also extends to the other main concepts of the paper: With respect to self-knowledge, I would like to understand more as a reader what "knowing" usually implies, and especially what we know the retention and encoding of knowledge in LLM. Here I would refer for instance to the excellent work of [3].
    * W.r.t. self-expression, I find the distinction very useful and educational, but the definition a bit disappointing. Given the amount of cited works in section 2.3, I wish the authors would elaborate more on the insights gained in these surveyed works, namely, given some definition of knowledge from an improved previous section, what conditions need to be in place for a model to express knowledge consistently, and under which condition it might fail. Can we say with certainty that something is known if it cannot consistently be expressed?
     * A related aspect to this problem for me seems to be that the evaluation of self-knowledge and self-expressions seems to be entangled; i.e. since we typically seem to evaluate the presence of knowledge in a model through questions, we only detect known knowledge that is expressed faithfully. How do we know that knowledge is present but not expressed consistently, and could there be cases where incomplete or missing knowledge is being expressed faithfully by chance or circumstancially? Can we detect knowledge in models without explicitly asking a question (thinking for instance about works on knowledge editing)? I think a discussion around these issues would improve the survey a lot.
* The authors seem to have detected contradictory definitions of some metrics like the refusal rate in equation 2 or the performance spread in equation 7. I would appreciate if the authors could reflect more on the impact of these subtle difference and give these measures slightly different names, so that subsequent works can refer to this survey for an unambiguous definition.
* There are more alternatives to the ECE that are not mentioned in the paper, specifically [4-10].
* In the summary & discussion paragraph 7, the sentence "The task of recognizing known and unknown requires the model to identify what it knows and what it doesn't" reads a bit superfluous and could maybe be clarified or removed.
* The concept of performance spread in figure 4 does not appear very intuitive, maybe the example could be expanded with more answers (and maybe the definition of the metric?) to explain where the result 1 comes from.
* Regarding the discussions on p. 11, I would like to add some thoughts. For one, while language models might achieve good calibration on a token level after seeing sufficient amounts of data, there is no inherent reason in the training pipeline of modern language models why the confidence derived from a sequence of tokens should reflect *correctness* - as shown in equation 10, it is simply the product of token probabilities. Secondly, calibration is a bit tricky to assess even on a token-level since there might not be a single ground truth when for instance the next predicted word could be one of a set of synonyms. Thirdly, for the same reason, token probabilities do not capture (only) model confidence but general aleatoric uncertainty in the choice of the next word. For reference, see e.g. [11]. This is why I mostly disagree with the conclusion in the following discussion paragraph that a) token-level calibration is strong and that b) token-level calibration is indicative of a potential good calibration on the sequence-level, based on token-level confidence scores.
* There are additional references that show the shortcomings of verbalized uncertainty, i.e. [12, 13].
* The work of Ulmer et al. on page 13 is slightly mischaracterized, in that the authors do not finetune the target LLM, but an additional model, and the described method is used to create labels for this additional model. The work could be grouped with others that try to predict the confidence of a model directly by training a secondary model, including [14, 15].
* I would like to see a conclusion that is just a little bit more expanded in order to better reflect some of the key takeaways of the paper, for instance the distinction into self-knowledge and self-expression and some very rough sketch of the current state and challenges of both.
* Other minor changes
    * Figure 3 is located far away from its first mention.
    * Typo p.4 bottom "of the model's known knowledge as it is othen included"
    * Use $F_1$ for F1-score.
    * Inconsistent paragraph beginnings: In some sections the "Summary & Discussion" paragraph begins in bold, while in others it's in italics.
    * Usage of \citealp or similar: Some spots in the paper using double brackets instead of parentheses and \citealp, for instance on the bottom of p.4 "(e.g., SQuAD (Rajpurkar et al., 2016))".

[1] Dhuliawala, Shehzaad, Vilém Zouhar, Mennatallah El-Assady, and Mrinmaya Sachan. "A Diachronic Perspective on User Trust in AI under Uncertainty." arXiv preprint arXiv:2310.13544 (2023).

[2] Zhou, Kaitlyn, Jena D. Hwang, Xiang Ren, and Maarten Sap. "Relying on the Unreliable: The Impact of Language Models' Reluctance to Express Uncertainty." arXiv preprint arXiv:2401.06730 (2024).

[3] Fierro, Constanza, Ruchira Dhar, Filippos Stamatiou, Nicolas Garneau, and Anders Søgaard. "Defining knowledge: Bridging epistemology and large language models." arXiv preprint arXiv:2410.02499 (2024).

[4] Kumar, Ananya, Percy S. Liang, and Tengyu Ma. "Verified uncertainty calibration." Advances in Neural Information Processing Systems 32 (2019).

[5] Zhang, Jize, Bhavya Kailkhura, and T. Yong-Jin Han. "Mix-n-match: Ensemble and compositional methods for uncertainty calibration in deep learning." In International conference on machine learning, pp. 11117-11128. PMLR, 2020.

[6] Gruber, Sebastian, and Florian Buettner. "Better uncertainty calibration via proper scores for classification and beyond." Advances in Neural Information Processing Systems 35 (2022): 8618-8632.

[7] Kirchenbauer, John, Jacob Oaks, and Eric Heim. "What is Your Metric Telling You? Evaluating Classifier Calibration under Context-Specific Definitions of Reliability." arXiv preprint arXiv:2205.11454 (2022).

[8] Roelofs, Rebecca, Nicholas Cain, Jonathon Shlens, and Michael C. Mozer. "Mitigating bias in calibration error estimation." In International Conference on Artificial Intelligence and Statistics, pp. 4036-4054. PMLR, 2022.

[9] Błasiok, Jarosław, and Preetum Nakkiran. "Smooth ECE: Principled reliability diagrams via kernel smoothing." arXiv preprint arXiv:2309.12236 (2023).

[10] Chidambaram, Muthu, Holden Lee, Colin McSwiggen, and Semon Rezchikov. "How Flawed is ECE? An Analysis via Logit Smoothing." arXiv preprint arXiv:2402.10046 (2024).

[11] Ilia, Evgenia, and Wilker Aziz. "Predict the Next Word:< Humans Exhibit Uncertainty in this Task and Language Models _>." In Proceedings of the 18th Conference of the European Chapter of the Association for Computational Linguistics, vol. 2, pp. 234-255. 2024.

[12] Yona, Gal, Roee Aharoni, and Mor Geva. "Can Large Language Models Faithfully Express Their Intrinsic Uncertainty in Words?." arXiv preprint arXiv:2405.16908 (2024).

[13] Krause, Lea, Wondimagegnhue Tufa, Selene Báez Santamaría, Angel Daza, Urja Khurana, and Piek Vossen. "Confidently wrong: exploring the calibration and expression of (Un) certainty of large language models in a multilingual setting." In Proceedings of the workshop on multimodal, multilingual natural language generation and multilingual WebNLG Challenge (MM-NLG 2023), pp. 1-9. 2023.

[14] Fathullah, Yassir, Puria Radmard, Adian Liusie, and Mark JF Gales. "Who Needs Decoders? Efficient Estimation of Sequence-level Attributes." arXiv preprint arXiv:2305.05098 (2023).

[15] Liu, Linyu, Yu Pan, Xiaocheng Li, and Guanting Chen. "Uncertainty Estimation and Quantification for LLMs: A Simple Supervised Approach." arXiv preprint arXiv:2404.15993 (2024).

**Strengths And Weaknesses:**

Strengths
-----------

* The paper is easy to read
* The paper is well-structured
* The paper gives a broad overview over several important lines of work from a fresh angle.

Weaknesses
--------------

* The core concepts, namely honesty, (self-)knowledge and self-expression are only superficially defined. Especially for a computer science audience, I believe the unique angle of this paper could be to give better definitions from the social sciences for these terms before applying them in an LLM context (see requested changes).
* Some definitions of other concepts seem vague or overlapping (see requested changes)

---

> ### Author Response · Authors · 2025-01-22
> **Response to Reviewer eGs9 (Part 1)**
>
> Thank you for your thoughtful review and constructive feedback. We appreciate your positive comments on the paper's structure and readability. And we have provided point-to-point changes to address your concerns as follows:
>
> > **Requested Change 1.1: I like that the paper surveys the field from the perspective of honesty. However, I find honesty and the notions of self-knowledge and self-expression only superficially defined. Honesty for instance is only given as a dictionary definition. But how is honest interaction defined betweem humans? How does human interaction benefit from honesty? How and when does it suffer from dishonesty? Which of these insights are applicable to human-AI interactions? How does it relate to trust? Especially when thinking of the last two questions, I think of the works of [1, 2], but especially with the rest I would like to learn more about the definition and insights from fields such as e.g. philosophy, psychology, and game theory. Not only do I think it would set the survey apart and would be very educational, but also set the survey on a solid foundation.**
>
> Thank you for your valuable feedback regarding the definition and conceptualization of honesty. We appreciate your suggestion to deepen the discussion and have revised Section 2 of the paper accordingly, with the changes highlighted in blue.
>
> You can see the revised content at **the beginning of Section 2**. We have added two new paragraphs. The first discusses the significance of honesty in human society, while the second explores its importance in human-AI interaction. This addition should help readers better understand the crucial role of honesty.
>
> ---
>
> > **Requested Change 1.2: This critique also extends to the other main concepts of the paper: With respect to self-knowledge, I would like to understand more as a reader what "knowing" usually implies, and especially what we know the retention and encoding of knowledge in LLM. Here I would refer for instance to the excellent work of [3].**
>
> Thank you for your insightful comments. We have expanded the discussion in Section 2.2 in the updated version of the paper, with the revisions highlighted in blue (**footnote 2 on page 4**).
>
> In Section 2.2, when discussing “know,” we have referred to the widely recognized usage of “know” in LLMs and clarified its meaning in the context of this paper. Additionally, we have provided relevant references on the mechanisms of knowledge in LLMs for readers to explore further.

---

> ### Author Response · Authors · 2025-01-22
> **Response to Reviewer eGs9 (Part 2)**
>
> > **Requested Change 1.3: W.r.t. self-expression, I find the distinction very useful and educational, but the definition a bit disappointing. Given the amount of cited works in section 2.3, I wish the authors would elaborate more on the insights gained in these surveyed works, namely, given some definition of knowledge from an improved previous section, what conditions need to be in place for a model to express knowledge consistently, and under which condition it might fail. Can we say with certainty that something is known if it cannot consistently be expressed?**
>
> Thank you for your insightful comments regarding the definition of self-expression. We have revised **Section 2.3** in the updated version of the paper to address your concerns, and the changes are highlighted in blue.
>
> Specifically, we have further elaborated on the following points:
>
> 1. In the first paragraph of Section 2.3, we further discuss why research on self-expression is important.
> 2. In the second and third paragraphs of Section 2.3, we provide a more detailed discussion of various works related to self-expression and offer explanations for the deficiencies in self-expression abilities, such as the lack of necessary data augmentation or the introduction of inappropriate training data during the fine-tuning stage. For issues that remain unresolved, we prefer not to provide potentially confusing explanations. We will continue to monitor research on these issues and update this work accordingly.
> 3. Regarding the question, “Can we say with certainty that something is known if it cannot consistently be expressed?” we believe this question is more relevant to the field of epistemology. In the context of computer science, we are more focused on specific performance metrics. Therefore, we do not provide an answer to this question in the paper to avoid confusion. Personally, we believe that even if a piece of knowledge cannot be consistently expressed correctly, the model may still “know” it, but lacks an appropriate mechanism for expression. For example, in the Reversal Curse [1], even if an LLM knows “A is B,” it may fail to infer “B is A.” We see this as a limitation in the model’s self-expression abilities and hope that future research will address this issue.
>
> [1] Berglund, L., Tong, M., Kaufmann, M., Balesni, M., Stickland, A. C., Korbak, T., & Evans, O. The Reversal Curse: LLMs trained on “A is B” fail to learn “B is A”. In The Twelfth International Conference on Learning Representations.
>
> ---
>
> > **Requested Change 1.4: A related aspect to this problem for me seems to be that the evaluation of self-knowledge and self-expressions seems to be entangled; i.e. since we typically seem to evaluate the presence of knowledge in a model through questions, we only detect known knowledge that is expressed faithfully. How do we know that knowledge is present but not expressed consistently, and could there be cases where incomplete or missing knowledge is being expressed faithfully by chance or circumstancially? Can we detect knowledge in models without explicitly asking a question (thinking for instance about works on knowledge editing)? I think a discussion around these issues would improve the survey a lot.**
>
> Thank you very much for your valuable feedback. The issue you raised regarding the evaluation of self-knowledge and self-expression is indeed worthy of in-depth discussion. Below, we would like to address your suggestions from the perspectives of **knowledge identification**:
>
> - In Section 3.1 and Section 3.2, we mention that evaluations rely on knowledge identification to construct test datasets (e.g., Recognition of Known/Unknown in Section 3.1 and the Identification-based Evaluation in Section 3.2 ). Then, based on the identified knowledge, self-knowledge capability involves the model being aware of its own knowledge and then responding with whether it knows this knowledge (as illustrated in Figure 3). Meanwhile, self-expression capability refers to the model’s ability to faithfully express this knowledge (as shown in Figure 4).
> - As you rightly pointed out, current knowledge identification approaches may have limitations: we might fail to detect knowledge that exists internally but is not consistently expressed, or in some cases, the model might faithfully express incomplete or missing knowledge by chance. In response to this concern, we have discussed the limitations of current knowledge identification methods in Section 6 (Future Work), particularly their reliance on the model's external expressions. We propose that future research could explore identifying knowledge internalized within the model, such as by analyzing knowledge embedded in the model's parameters or present in the context.
>
> We have included these discussions in the revised version in the **“Knowledge Identification” part of Section 6** , which are highlighted in blue. Once again, thank you for your insightful comments and suggestions.

---

> ### Author Response · Authors · 2025-01-22
> **Response to Reviewer eGs9 (Part 3)**
>
> > **Requested Change 2: The authors seem to have detected contradictory definitions of some metrics like the refusal rate in equation 2 or the performance spread in equation 7. I would appreciate if the authors could reflect more on the impact of these subtle difference and give these measures slightly different names, so that subsequent works can refer to this survey for an unambiguous definition.**
>
> We appreciate your constructive feedback regarding the refinement of our metric definitions. The **Refusal Rate** and **Performance Spread** metrics are based on previous research, and their calculations exhibit slight variations due to differing emphases across studies. We have provided a detailed discussion of these distinctions to ensure clarity. Additionally, we have introduced subscripts to the original metric names to facilitate unambiguous referencing in subsequent research.
>
> (1) **Refusal Rate** emphasizes the model's ability to recognize unknowns, measuring the percentage of cases in which the model correctly refuses to respond. It is divided into two metrics based on the ground-truth nature of the questions:
>
> - For ground-truth known questions, the refusal rate ( $\text{Refusal Rate}_{\text{known}}$ ) measures the model's tendency to incorrectly refuse to respond. A lower value is desired, indicating fewer unnecessary refusals:
>
>     $$
>     \text{Refusal Rate}_{\text{known}} = \frac{N_3}{N_1 + N_3}.
>     $$
>
> - For ground-truth unknown questions, the refusal rate ( $\text{Refusal Rate}_{\text{unknown}}$ ) measures the model's ability to correctly refuse to respond. A higher value is preferable, reflecting better recognition of unknowns:
>
>
>     $$
>     \text{Refusal Rate}_{\text{unknown}} = \frac{N_4}{N_2 + N_4}.
>     $$
>
>
> (2)  **Performance Spread** measures the performance gap among the augmented examples and is mainly used in the context of the format adjustment and instruction rephrasing strategy. Depending on whether the spread is compared against the minimum or average performance, two variants are defined:
>
> $$
> \text{Performance Spread}_{\text{max-min}} = \text{maxP}(X) - \text{minP}(X),
> $$
>
> $$
> \text{Performance Spread}_{\text{max-avg}} = \text{maxP}(X) - \text{avgP}(X),
> $$
> where $ X $ represents the augmented example dataset, while $\text{maxP}(\cdot), \text{minP}(\cdot)$ and $\text{avgP}(\cdot)$ denote the operation to get the maximum, minimum and average performance from $X$.
>
> We have incorporated these discussions in the **“Recognition of Known/Unknown” part of Section 3.1** and **“Identification-free Evaluation” part of Section 3.2** of the revised manuscript, which are highlighted in blue. Once again, we thank you for your valuable suggestion.
>
> ---
>
> > **Requested Change 3: There are more alternatives to the ECE that are not mentioned in the paper, specifically [4-10].**
>
> We are grateful for your insightful suggestion to include additional alternatives to the Expected Calibration Error (ECE).
>
> In the **“Calibration” part of Section 3.1**, we have expanded the discussion to include these alternatives, with the revisions highlighted in blue. Specifically, we have summarized the limitations of the original ECE and introduced additional alternatives to address these issues, supported by relevant references. We encourage future studies to consider strengths and limitations of these alternatives and adopt the most suitable metrics.
>
> ---
>
> > **Requested Change 4: In the summary & discussion paragraph 7, the sentence "The task of recognizing known and unknown requires the model to identify what it knows and what it doesn't" reads a bit superfluous and could maybe be clarified or removed.**
>
> Thank you for your suggestion regarding the statement. We have revised this sentence to “One line of research investigates the LLMs' capacity to make binary judgments on the recognition of known/unknown task” in the **“Summary & Discussion” part of Section 3.1**, highlighted in blue.
>
> ---
>
> > **Requested Change 5: The concept of performance spread in figure 4 does not appear very intuitive, maybe the example could be expanded with more answers (and maybe the definition of the metric?) to explain where the result 1 comes from.**
>
> Thank you for your suggestion regarding the clarity of performance spread and the illustration in Figure 4. As mentioned in our previous response, we have provided the definition of performance spread. Furthermore, we have revised **Figure 4** in the updated manuscript to enhance its clarity. We hope this resolves any confusion. We greatly appreciate your constructive feedback.

---

> ### Author Response · Authors · 2025-01-22
> **Response to Reviewer eGs9 (Part 4)**
>
> > **Requested Change 6: Regarding the discussions on p. 11, I would like to add some thoughts. For one, while language models might achieve good calibration on a token level after seeing sufficient amounts of data, there is no inherent reason in the training pipeline of modern language models why the confidence derived from a sequence of tokens should reflect *correctness* - as shown in equation 10, it is simply the product of token probabilities. Secondly, calibration is a bit tricky to assess even on a token-level since there might not be a single ground truth when for instance the next predicted word could be one of a set of synonyms. Thirdly, for the same reason, token probabilities do not capture (only) model confidence but general aleatoric uncertainty in the choice of the next word. For reference, see e.g. [11]. This is why I mostly disagree with the conclusion in the following discussion paragraph that a) token-level calibration is strong and that b) token-level calibration is indicative of a potential good calibration on the sequence-level, based on token-level confidence scores.**
>
> Thank you for your thoughtful comments regarding token-level calibration. In response, we have revised the **“Summary & Discussion” part of Predictive Probability in Section 4.1,** in the updated version of the paper to address these points, highlighted in blue.
>
> Specifically, we have further clarified and elaborated on the following aspects:
>
> 1. We have revised the previous explanation, clarifying that the well-calibrated token-level prediction is reflected in constrained token-level tasks.
> 2. We have discussed in more detail the reasons for the poorer performance of sequence-level calibration.
> 3. We have also explored potential improvements to address these issues.
>
> ---
>
> > **Requested Change 7: There are additional references that show the shortcomings of verbalized uncertainty, i.e. [12, 13].**
>
> Thank you for your valuable comments on the shortcomings of verbalized uncertainty. In the updated version of our paper, we have incorporated the discussion of these papers into **Section 4.1, the part on Prompting**, highlighted in blue.
>
> ---
>
> > **Requested Change 8: The work of Ulmer et al. on page 13 is slightly mischaracterized, in that the authors do not finetune the target LLM, but an additional model, and the described method is used to create labels for this additional model. The work could be grouped with others that try to predict the confidence of a model directly by training a secondary model, including [14, 15].**
>
> Thank you for your understanding and clarification regarding the work of Ulmer et al. In the updated version of our paper, we have revised the description of this work, and group it with the others that try to predict confidence with another model. The details can be found in **Section 4.2, the part on Supervised Fine-tuning** in the updated version of this paper, highlighted in blue.
>
> ---
>
> > **Requested Change 9: I would like to see a conclusion that is just a little bit more expanded in order to better reflect some of the key takeaways of the paper, for instance the distinction into self-knowledge and self-expression and some very rough sketch of the current state and challenges of both.**
>
> Thank you very much for your suggestion. We have incorporated the revisions in **Section 7** of the updated version, with the changes highlighted in blue. Specifically, we have:
> 1.	Clarified the distinction between self-knowledge and self-expression.
> 2.	Discussed the current state and key future directions.

---

> ### Author Response · Authors · 2025-01-22
> **Response to Reviewer eGs9 (Part 5)**
>
> > **Requested Change 10: Other minor changes**
>
> Thank you for your suggestions to improve the writing details of our paper. In the updated version, we have addressed these points with changes highlighted in blue:
>
> | No. | Comment                                                                 | Response                                                                                       |
> |-----|-------------------------------------------------------------------------|-----------------------------------------------------------------------------------------------|
> | 1   | Figure 3 is located far away from its first mention.                     | We have relocated Figure 3 to page 5.                                                         |
> | 2   | Typo p.4 bottom "of the model's known knowledge as it is othen included” | We have changed the word “othen” to “often” in the “Recognition of Known/Unknown” part of Section 3.1. |
> | 3   | Use F1 for F1-score.                                                     | We have changed “F1” to “$F_1$” in the “Recognition of Known/Unknown” part of Section 3.1 and Figure 3. |
> | 4   | Inconsistent paragraph beginnings: In some sections the "Summary & Discussion" paragraph begins in bold, while in others it's in italics. | We have changed all "Summary & Discussion" paragraphs to begin in italics in Section 3.       |
> | 5   | Usage of \citealp or similar: Some spots in the paper using double brackets instead of parentheses and \citealp, for instance on the bottom of p.4 "(e.g., SQuAD (Rajpurkar et al., 2016))". | We have fixed the issue of double brackets in citations in “Recognition of Known/Unknown” part of Section 3.1, “Identification-based Evaluation” part of Section 3.2, and “Honesty in Various Models” part of Section 6. |

---

> ### Comment · Reviewer_eGs9 · 2025-01-23
> **Comment to Author Response**
>
> I thank the authors for their detailed response in addressing all of my criticisms.
> I am happy to see the paper improved even further, and think that it will be a valuable resource for the community.
>
> Last little comment: When reviewing the changes, it appears that the authors forgot a fullstop right before the reference to footnote 4 on page 4.

---

> > ### Author Response · Authors · 2025-01-23
> > **Response to Reviewer eGs9 (Part 6)**
> >
> > Thank you very much for your recognition of our work. Your detailed and thoughtful feedback has greatly helped us improve the quality of the paper. We hope this work will contribute positively to the LLM research community.
> >
> > Regarding the issue with the fullstop, we have made the necessary revisions and have uploaded the updated version of the paper.

---

### Author Response · Authors · 2025-01-22
**Appreciation for Reviewers’ Feedback and AE’s Support**

We would like to express our sincere gratitude to all reviewers for their constructive and insightful feedback, which has significantly enhanced the quality of our paper. We also thank the action editor for attentive follow-up throughout the review process.

The revised manuscript has been uploaded, with changes from the original submission highlighted in blue.

Should there be any further questions or suggestions, we warmly welcome continued discussion and are eager to engage further.

---

### Decision · Action_Editor_gLCs · 2025-03-08

**Recommendation:** Accept as is

**Comment:**

The paper breaks down the LLM honesty definitions and evaluations into two broad sub-categories: self-knowledge and self-expression. The first deals with models internal understanding of the limits of its knowledge, and the second deals with the model being able to consistently answer questions about what it knows. For each of these fields, the authors meticulously survey the literature on both definitions and evaluations, providing a unified perspective. The last part of the paper deals with methods focused on improving honesty: prompting, using token-level uncertainty, supervised training, reinforcement learning, etc.

This is a good and valuable paper about a new fast-developing field. It will be a valuable resource for the researchers who want to get a clear understanding of the state of the field, its methods and definitions.

The reviewers were also unanimous in supporting the paper. All of the reviewers indicated that the paper is strong, and that all of their concerns were addressed during the discussion phase.

**Audience:**

Honesty in language models is an important topic in modern LLM research. The paper will be a valuable resource for many researchers looking to better understand this field.

**Claims And Evidence:**

This is a survey paper that reviews and provides a unified perspective on the field of language model honesty. The authors cover a wide range of results, methods and ideas in the filed. The claims in the paper are well-supported.